# Parietal alpha frequency shapes own-body perception by modulating the temporal integration of bodily signals

Mariano D'Angelo [1] ✉, Renzo C. Lanfranco [1,2], Marie Chancel[3] & H. Henrik Ehrsson [1]

An influential proposal in the field of cognitive neuroscience suggests that alpha-frequency brain oscillations constrain the temporal sampling of external sensory signals, shaping the temporal binding window (TBW)—the interval during which sensory signals are integrated. However, whether alpha frequency modulates the integration of self-related sensory signals and the perception of the body as one's own (body ownership) remains unknown. Here, we demonstrate that individual alpha frequency (IAF) from the parietal cortex predicted TBWs and perceptual sensitivities in body ownership and visuotactile simultaneity judgment tasks, with faster frequencies narrowing TBWs and increasing sensitivities, and vice versa. Modulating IAF through brain stimulation altered TBWs and sensitivities, establishing a causal relationship. Computational modeling linked IAF to uncertainty in asynchrony information within the causal inference process. These findings demonstrate that parietal alpha frequency shapes the sense of body ownership by modulating the temporal integration of bodily sensory signals.

Multisensory integration refers to the brain's ability to combine sensory information from different modalities into coherent perceptions[1]. This integration process optimizes perception by increasing precision and reducing uncertainty through the utilization of all available sensory information. However, a fundamental challenge for the brain, discussed in the 19th century by the pioneering physiologist and physicist Hermann von Helmholtz, is determining which sensory signals should be integrated and which should be segregated. Sensory signals originating from the same external object or event should be integrated, whereas those from different sources should not. Information from the spatiotemporal correlations among sensory signals helps solve this 'multisensory integration problem', with temporal discrepancies playing a critical role[2]. When there is no, or only a very small, temporal discrepancy between two different signals, they are more likely to be caused by the same event or object and should therefore be integrated. Conversely, larger temporal discrepancies suggest that the signals relate to different events or objects and thus

that multisensory integration should not occur, leaving the signals segregated. Therefore, the 'temporal binding window' (TBW) is a crucial notion in multisensory integration. The TBW refers to the temporal interval within which sensory stimuli are more likely to be integrated into a unified percept[3–7]. The TBW also relates to the ability to discriminate different events in time, which depends on signals being separated outside the binding window. The TBW varies across individuals and contributes to individual differences in the temporal resolution of multisensory information processing[3–7].

An influential theory in psychology and neuroscience research proposes that alpha oscillations play a role in sampling sensory signals into discrete perceptual frames, thereby determining the width of the TBW[8–13]. According to this theory, visual perception and multisensory perception occur within a discrete temporal window governed by the frequency of alpha oscillations, which segment perceptual information into perceptual units. Consequently, variations in the frequency of alpha oscillations predict variations in the TBW and the temporal

[1]Department of Neuroscience, Karolinska Institutet, Stockholm, Sweden. [2]Department of Clinical Neuroscience, Karolinska Institutet, Stockholm, Sweden. [3]Aix Marseille University, CNRS, CRPN (Centre for Research in Psychology and Neuroscience- UMR 7077), Marseille, France. ✉e-mail: mariano.dangelo@ki.se

resolution of perception. Recent studies support the idea that higher frequencies result in more frequent perceptual frames, increasing the probability of segregating two stimuli over time and resulting in a narrower TBW. Conversely, slower oscillations generate fewer perceptual units and a wider TBW, which increases the likelihood of temporal integration[14–20]. However, a recent study[21] questioned whether alpha frequency plays a role in the temporal resolution of perception based on a null finding, sparking debate[22–27] and emphasizing the need for additional experiments using state-of-the-art psychophysics and EEG analysis methods to further test this theory[27].

In addition to combining or segregating sensory signals from external stimuli, the brain must determine whether to integrate or segregate signals originating from one's own body. This is particularly important for the sense of body ownership—the perceptual experience of one's limbs and body parts as one's own[28–30]. Body ownership depends on the multisensory integration of self-related signals from different sensory modalities, including vision, touch, and proprioception. Temporal discrepancies among these signals are critical in determining whether they should be integrated and whether body ownership is perceived[29–33] This integration is mediated by neuronal populations in the posterior parietal and premotor cortices, where bodily sensory signals converge[34–41]. The posterior parietal cortex specifically plays a key role in determining whether visuotactile signals should be combined or segregated in the process of body ownership perception[39]. However, this understanding is derived primarily from functional magnetic resonance imaging studies[37–39] and electrophysiological recordings of multiunit and local high gamma activity[40,42], which focus on signal intensity in localized areas. Such studies highlight neuronal activity in small patches of the cortex but do not reveal how brain oscillations modulate the temporal integration of information across underlying local neural networks. Additionally, little is known about the electrophysiological mechanisms determining the temporal resolution of self-related multisensory information processing that underpins the sense of body ownership.

Here, we hypothesized that individual alpha frequency (IAF) drives the temporal integration of bodily sensory information responsible for body ownership, similar to how these oscillations are believed to support the temporal integration of visual and auditory signals for external events. To test this hypothesis, we used a psychophysical version of the rubber hand illusion[31,39] together with electroencephalography (EEG) and brain stimulation. In the rubber hand illusion, participants experience synchronous touches on a rubber hand in view and their real hand, which is hidden. After a short stimulation period, most participants perceive the touches as originating from the rubber hand, which begins to feel like their own and part of their body[43–45]. In our psychophysical setup, we manipulated the timing of taps on the rubber hand relative to the participants' real hand, introducing delays or advances up to 500 ms, and asked participants to judge illusory hand ownership after each trial in a detection-like task. As longer delays gradually weaken the illusory feeling of hand ownership, this paradigm enabled us to map a fine-grained relationship between delay and ownership judgment tasks, quantifying the TBW of body ownership and perceptual sensitivity using signal detection theory (SDT)[31]. By comparing these measures to EEG recordings from the same subjects, we could investigate the role of alpha oscillations in the temporal integration of sensory information that drives body ownership perception. In all experiments, we also included a visuotactile simultaneity judgment task to directly test the temporal resolution of multisensory perception, in which participants had to judge whether brief tactile and external visual stimuli were simultaneous or delayed.

In Experiment 1, we found that the TBW and sensitivity to visuotactile simultaneity correlated with the TBW and sensitivity to body ownership, as would be expected if IAF serves as a common electrophysiological mechanism for the temporal integration of visual and somatosensory signals. In Experiment 2, we found evidence that IAF predicts participants' TBW and sensitivity to body ownership and visuotactile simultaneity. In Experiment 3, we examined the causal role of parietal alpha frequency in shaping TBW and sensitivity related to body ownership and visuotactile simultaneity. This was achieved using brain stimulation (transcranial alternating current stimulation–tACS) to either increase or decrease participants' cortical alpha frequency while observing the predicted changes in TBW and sensitivity. Finally, to investigate the computational mechanism, we fitted our behavioral data from Experiments 2 and 3 to a previously validated Bayesian causal inference model of body ownership[29] and found that IAF was related to sensory uncertainty; this finding suggests that IAF influences perceptual inference by affecting the reliability of multisensory asynchrony information. Collectively, our psychophysical, EEG, brain stimulation, and computational modeling results provide conclusive evidence that alpha frequencies play a crucial role in mediating the temporal integration of visual and tactile signals in body ownership, determining the temporal resolution of self-related multisensory information processing.

## Results

### Experiment 1: correlation between body ownership and visuotactile simultaneity TBWs

Experiment 1 clarified whether the temporal resolution of visuotactile simultaneity correlates with the temporal resolution of multisensory integration involved in body ownership. To this end, participants performed two psychophysical detection tasks: *(i)* a body ownership judgment task to evaluate the temporal resolution of multisensory integration in body ownership and *(ii)* a simultaneity judgment task to assess the temporal resolution of the simultaneity perception of external visuotactile stimuli.

During body ownership judgment tasks, we used the rubber hand illusion[29,41] to create the feeling that a prosthetic hand belonged to the participant's own body. To create this illusion, two robotic arms applied six repetitive tactile stimuli to the index finger of the rubber hand in view and to the participant's hidden real index finger[29] (Fig. 1a). When the touches on the participant's hand and the rubber hand are synchronized, most participants report the perceptual sensation that the rubber hand feels like their own in the majority of trials[29,39]. In our setup, taps on the rubber hand were either synchronized with taps on the participant's real hand (synchronous condition) or delayed/advanced at four specific asynchronies up to 500 ms. In each trial, when the robots stopped, participants were instructed to verbally report whether the rubber hand felt like their own hand by saying "yes" ("the rubber hand felt like it was my hand") or "no" ("the rubber hand did not feel like it was my hand") (Fig. 1b).

In the simultaneity judgment task, participants judged the perceived simultaneity of two paired stimuli, i.e., a visual stimulus and a tactile stimulus (Fig. 1c). Tactile stimuli were delivered to the right index fingertip through a small vibrating motor. Visual stimuli consisted of a red-light-emitting diode. Visual and tactile stimuli were presented synchronously or with seven different asynchronies up to 500 ms (Fig. 1d).

These experimental setups allowed us to compute the TBW of both body ownership and simultaneity judgment tasks as the standard deviation of the Gaussian curve fitted on the "yes" responses as a function of the asynchronies[3–6,46]. We found that simultaneity TBW correlated with body ownership TBW (Pearson's $r = 0.664$; $p < 0.001$; Spearman's $\rho = 0.544$, $p = 0.002$) (Fig. 2a). In other words, participants who tolerated greater asynchronies and still reported perceptual synchrony also tolerated greater asynchronies when reporting experiencing illusory ownership of the rubber hand. Moreover, we observed that body ownership TBW was wider than simultaneity TBW ($t = 2,756$; $p = 0.010$; $d = 0.503$), as expected[29].

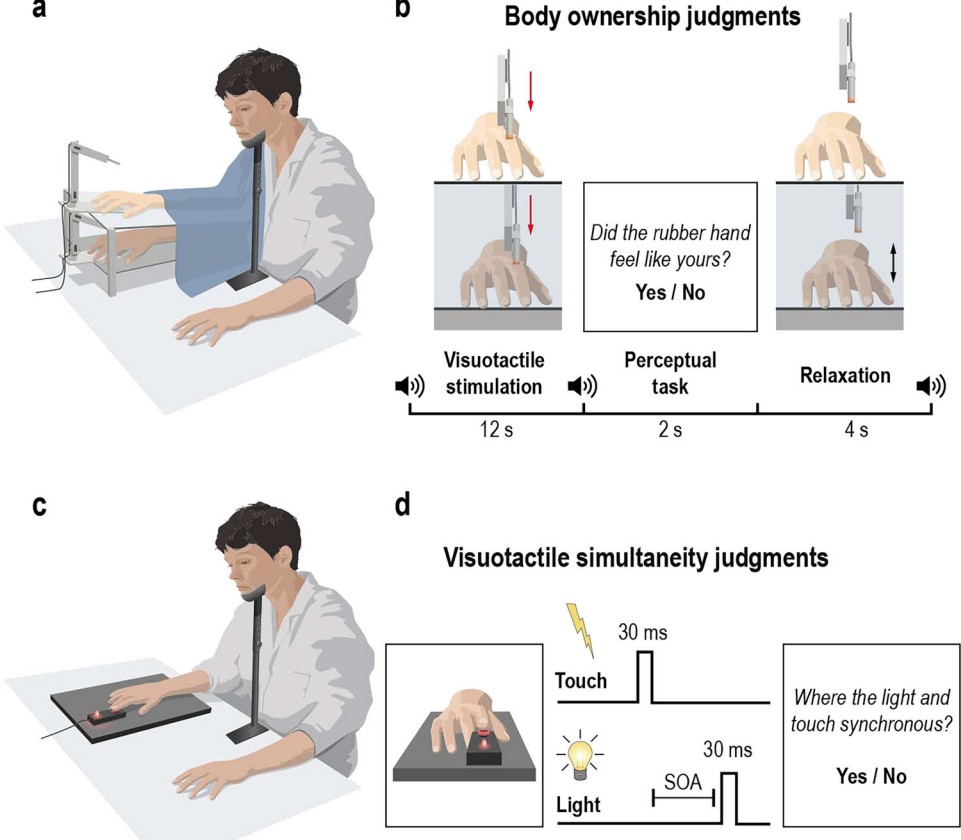

**Fig. 1 | Experimental setups and procedures in Experiment 1. a** Experimental setup for the body ownership judgment task. **b** Procedure for the body ownership judgment task. The participants' real right hand was hidden under a platform, while a right rubber hand was placed on top of the platform in their view. Both the rubber hand and the real hand were touched by robots six times over 12 s periods, either synchronously or with the rubber hand being touched slightly earlier or later. The degree of asynchrony was systematically manipulated between 0, ±50 ms, ±150 ms, ±300 ms, and ±500 ms. After each visuotactile stimulation period, the participants were asked to indicate whether the rubber hand felt like their own hand or not in a 'yes' or 'no' forced-choice task. The trials ended with a short relaxation period

before the next trials started. The participants were alerted to the different phases of the task with auditory cues (sound symbols). **c** Experimental setup for the simultaneity judgment task. **d** Procedure for the simultaneity judgment task. Participants judged the perceived simultaneity of visual (i.e., a red light) and tactile (i.e., a small vibration) stimuli. Each trial started with the presentation of either a visual or tactile stimulus for 30 ms, followed by the other stimulus after a variable asynchrony (stimulus onset asynchrony, SOA). The degree of asynchrony was systematically manipulated between 0 ms, ±50 ms, ±100 ms, ±150 ms, ±200 ms, ±300 ms, ±400 ms, and ±500 ms. In both tasks, head movement was minimized with a chinrest.

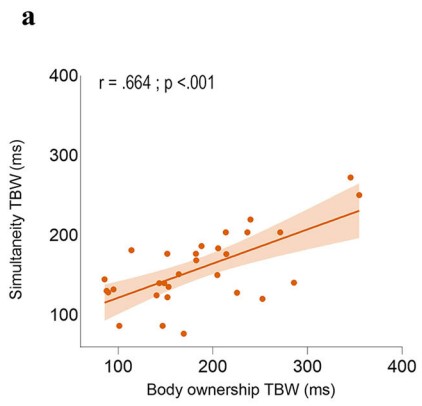

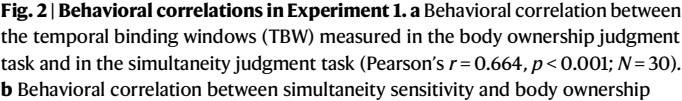

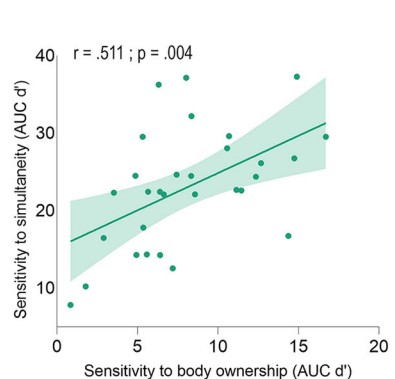

**Fig. 2 | Behavioral correlations in Experiment 1. a** Behavioral correlation between the temporal binding windows (TBW) measured in the body ownership judgment task and in the simultaneity judgment task (Pearson's $r = 0.664$, $p < 0.001$; $N = 30$). **b** Behavioral correlation between simultaneity sensitivity and body ownership

sensitivity (Pearson's $r = 0.551$; $p < 0.004$; $N = 30$). In both graphs, the solid lines represent the best-fitting regression. The shaded regions reflect the 95% confidence interval. The AUC denotes the area under the curve of the d' score measured for each asynchrony. Source data are provided as a Source Data file.

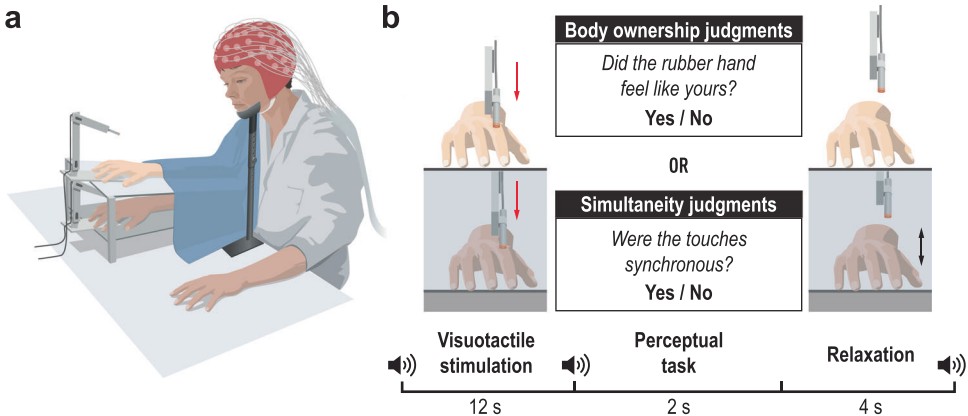

**Fig. 3 | Experimental setup and procedure in Experiment 2. a** Setup for the EEG experiment. **b** Procedure for body ownership and simultaneity judgment tasks. The visuotactile stimulation remained consistent across both tasks, with asynchronies varying across trials between 0 ms, ±100 ms, ±200 ms, ±300 ms, and ±400 ms.

## Experiment 1: correlation between body ownership and visuo-tactile sensitivities

The TBW measure may be susceptible to bias[21,47]. Therefore, we computed the sensitivity (d') of body ownership and simultaneity judgment tasks to each degree of asynchrony, i.e., the difference between the distribution of hits (i.e., reporting 'yes' when there was no visuotactile asynchrony; synchronous condition) and the distribution of false alarms (i.e., reporting 'yes' when the degree of visuotactile asynchrony was different than zero; asynchronous conditions). This SDT analysis aimed to measure sensitivity to simultaneity and to body ownership while controlling for possible bias[31,48]. Crucially, sensitivity to simultaneity judgment tasks significantly correlated with sensitivity to body ownership (Pearson's $r = 0.511$; $p = 0.004$; Spearman's $\rho = 0.529$; $p = 0.003$; Fig. 2b). This finding demonstrates that the correlation reflects a genuine relationship between body ownership and simultaneity perception, as we controlled for potential biases. Moreover, consistently with TBW results, sensitivity to body ownership was lower as compared to sensitivity to visuotactile simultaneity ($t = -12,44$; $p < 0.001$; $d = -2.272$). Both the sensitivity to body ownership ($F_{3,87} = 271.531$; $p < 0.001$; $\eta^2_p = 0.904$) and the sensitivity to simultaneity ($F_{6,174} = 368,761$; $p < 0.001$; $\eta^2_p = 0.927$) increased with increasing asynchronies, as expected[31].

## Experiment 2: correlations between IAF and TBWs

In Experiment 2, we tested whether IAF serves as the common electrophysiological mechanism underlying the temporal resolution of multisensory processing in simultaneity and body ownership perception. We focused on IAF in the posterior parietal cortex because previous fMRI studies have implicated this region in multisensory integration underlying body ownership[34–41] and parietal IAF is readily recorded during visuotactile paradigms[18]. Thus, we adopted a hypothesis-driven approach focusing on the PPC, where we hypothesized IAF to play a central role in visuotactile integration. We recorded IAF during both eyes-open resting-state and task conditions involving simultaneity and body ownership judgment tasks. Resting-state IAF is a neurophysiological trait that predicts participants' TBW[11,20,22]. Moreover, IAF can be modulated during task performance, facilitating either the integration or segregation of multisensory information depending on task demands[12,13,49]. Therefore, if IAF determines the temporal resolution of multisensory processing, as we hypothesized, we would expect both the resting-state and task-related IAFs to correlate with TBW and sensitivity to asynchrony.

Participants performed body ownership judgment tasks as in the previous experiment, along with a modified version of the simultaneity judgment task, while we recorded their brain activity using EEG (Fig. 3a). The modification was that the visuotactile stimulation was identical to that used in the body ownership judgment task; specifically, the robots tapped the rubber hand and participants' index fingers six times at the same intervals of asynchrony as in the body ownership task. This change was made to closely match the two tasks. At the end of the stimulation, the participants judged whether the touches applied to the prosthetic hand and their real hand were synchronous.

We estimated IAF using two methods. The first method is the classic approach, which identifies the local maximum in the power spectrum[10], a technique that is still commonly used[23]. However, due to its limitations (Supplementary Results of Experiment 2), we also applied the FOOOF algorithm[50,51] to compute IAF after controlling for aperiodic background activity in the EEG signal. Both methods yielded convergent results. Here, we present the results obtained using the classic method, while the FOOOF-IAF results are shown in the Supplementary Results of Experiment 2 and in Fig. S4.

As hypothesized, we observed negative correlations between the body ownership TBW and IAF during both resting-state and task conditions. Participants with slower resting-state IAF in both the left (Pearson's $r = -0.536$; $p < 0.001$; Spearman's $\rho = -0.589$ $p < 0.001$; Fig. S1a) and right (Pearson's $r = -0.579$; $p < 0.001$; Spearman's $\rho = -0.579$; $p = 0.001$) posterior parietal cortices exhibited a wider body ownership TBW. Similarly, slower parietal IAF recorded during body ownership judgment tasks was associated with a wider body ownership TBW in both the left (Pearson's $r = -0.651$; $p < 0.001$; Spearman's $\rho = -0.632$; $p < 0.001$, Fig. 4a) and right (Pearson's $r = -0.645$; $p < 0.001$; Spearman's $\rho = -0.661$; $p < 0.001$) hemispheres.

Resting-state IAF in the left (Pearson's $r = -0.519$; $p < 0.001$; Spearman's $\rho = -0.586$; $p < 0.001$; Fig. S1c) and right (Pearson's $r = -0.487$; $p = 0.001$; Spearman's $\rho = -0.488$, $p = 0.001$) parietal cortices also correlated with the simultaneity TBW. Additionally, task-related parietal IAF in both the left (Pearson's $r = -0.628$, $p < 0.001$; Spearman's $\rho = -0.636$, $p < 0.001$; Fig. 4c) and right (Pearson's $r = -0.597$, $p < 0.001$; Spearman's $\rho = -0.611$, $p < 0.001$) hemispheres correlated with the simultaneity TBW.

Notably, neither the body ownership nor simultaneity TBW correlated with alpha power, suggesting that TBW width is specifically related to the frequency, rather than the power, of alpha oscillations. In addition, given that bodily illusions are typically associated with a reduction in alpha power[52,53], we conducted a sanity check to confirm that alpha power differed between the resting state and the perceptual tasks (Supplementary Results for Experiment 2). Furthermore, the significant correlation between body ownership and simultaneity TBWs found in Experiment 1 was also observed in Experiment 2, again with a wider TBW for body ownership as compared to simultaneity (Supplementary Results for Experiment 2).

## Body ownership judgments

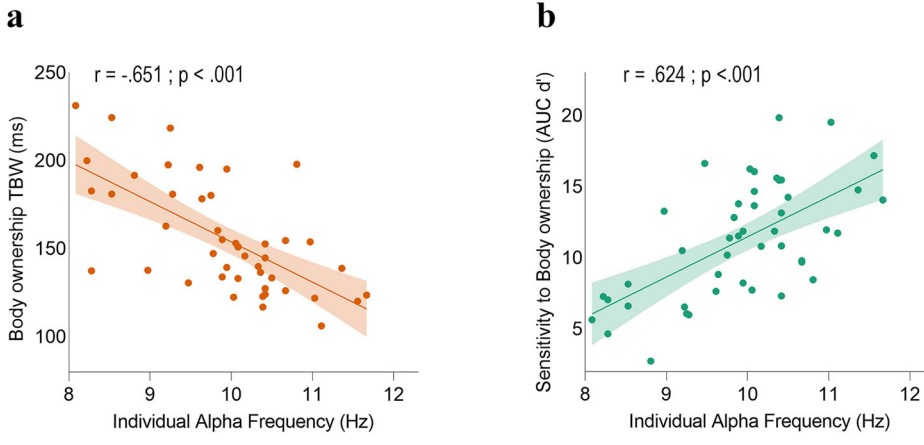

## Simultaneity judgments

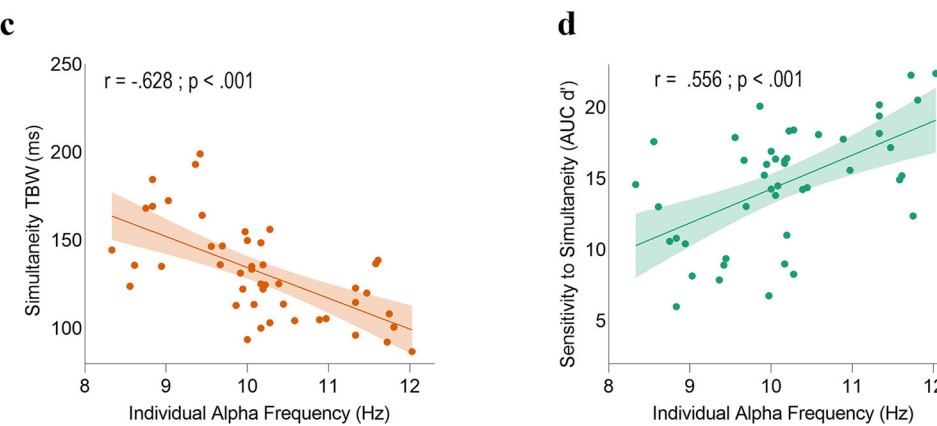

**Fig. 4 | Correlations between task-related parietal Individual Alpha Frequency (IAF) and TBW, as well as perceptual sensitivities, in Experiment 2. a** Correlation between task-related IAF measured in the left posterior parietal cortex (i.e., the hemisphere contralateral to the stimulation) during body ownership judgment tasks and body ownership TBW (Pearson's *r* = −.651, *p* < 0.001; *N* = 46). **c** Correlation between simultaneity task-related parietal IAF and visuotactile simultaneity TBW (Pearson's *r* = 0.628, *p* < 0.001; *N* = 46). **b** Correlation between body ownership task-related IAF in the left posterior parietal cortex and body ownership sensitivity (Pearson's *r* = 0.624, *p* < 0.001; *N* = 46). **d** Correlation between simultaneity task-related parietal IAF and visuotactile simultaneity sensitivity (Pearson's *r* = 0.556, *p* < 0.001; *N* = 46). In all graphs, the solid lines represent the best-fitting regression. The shaded regions reflect the 95% confidence interval. Source data are provided as a Source Data file. See Fig. S1 for correlations with resting-state IAF.

### Experiment 2: correlation between IAF and sensitivity

Next, using SDT analysis, we computed body ownership and simultaneity sensitivity, thereby controlling for potential bias. We found a significant correlation between body ownership sensitivity and resting-state IAF in both the left (Pearson's *r* = 0.471, *p* = 0.001; Spearman's *ρ* = 0.518, *p* < 0.001; Fig. S1b) and right (Pearson's r: 0.477, *p* = 0.001; Spearman's *ρ* = 0.473, *p* = 0.001) parietal lobes. The IAF registered during the body judgment tasks also significantly correlated with body ownership sensitivity in both the left (Pearson's *r* = 0.624; *p* < 0.001; Spearman's *ρ* = 0.594; *p* < 0.001; Fig. 4b) and right (Pearson's *r* = 0.583; *p* < 0.001; Spearman *ρ* = 0.654; *p* < 0.001) parietal cortices.

Similarly, the IAF recorded during both resting-state and the simultaneity tasks was correlated with simultaneity sensitivity. Participants with higher sensitivity displayed faster resting-state IAF in the left (Pearson's *r* = 0.420; *p* = 0.004; Spearman's *ρ* = 0.501, *p* < 0.001; Fig. S1d) and right (Pearson's *r* = 0.401, *p* = 0.006; Spearman's *ρ* = 0.431; *p* = 0.003) parietal cortices. Greater simultaneity sensitivity was also associated with faster task-related IAF in both the left (Pearson's r = 0.556; *p* < 0.001; Spearman's *ρ* = 0.525; *p* < 0.001; Fig. 4d) and

right (Pearson's *r* = 0.503, *p* < 0.001; Spearman's *ρ* = 0.491; *p* < 0.001) parietal cortices.

In contrast, there were no significant correlations of alpha power with body ownership or simultaneity sensitivities, nor between IAF and biases in body ownership or simultaneity judgment tasks (Supplementary Results for Experiment 2). Finally, we replicated two key behavioral findings from Experiment 1: the correlation between body ownership and simultaneity sensitivities, and the lower sensitivity observed for body ownership relative to simultaneity (Supplementary Results for Experiment 2).

### Experiment 2: post-hoc premotor IAF analyses

In a separate exploratory post-hoc analysis, we also analyzed frontal electrodes corresponding to the premotor cortex (see Supplementary Results for Experiment 2), another multisensory area strongly implicated in body ownership experience[34,40], to investigate whether the IAF PPC findings are unique to this area or generalize to other regions involved in body ownership. We found that both task-related and resting state premotor IAF from either hemisphere correlated significantly with body ownership sensitivity and body ownership TBW

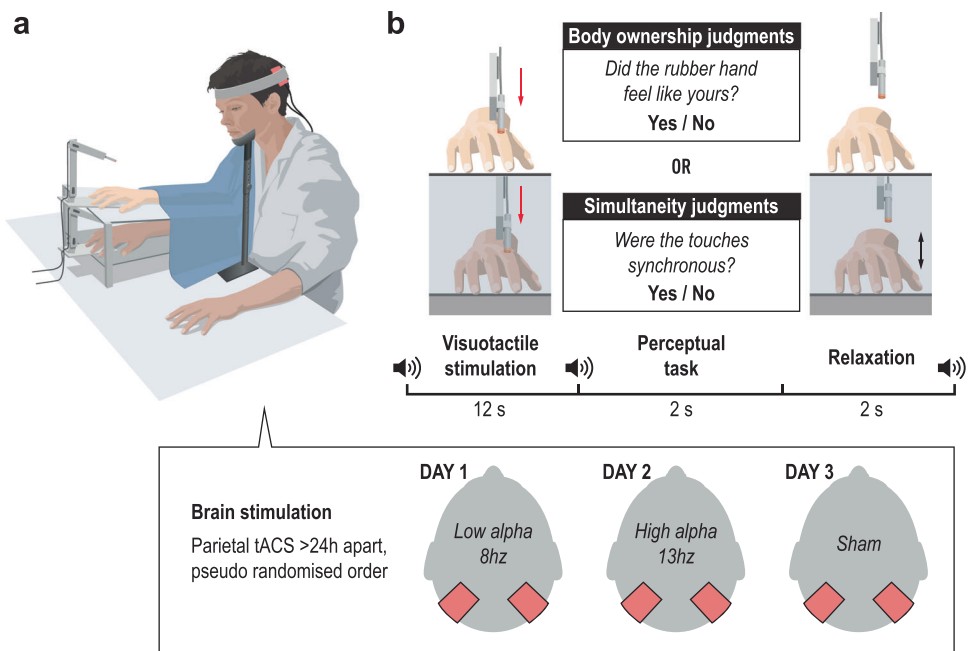

**Fig. 5 | Experimental setup and procedure in Experiment 3. a, b** Transcranial alternating current stimulation (tACS) experimental setup and procedure. Participants received continuous tACS over the posterior parietal cortex while performing body ownership and simultaneity judgment tasks in separate blocks. The visuotactile stimulation was identical for both tasks, with asynchronies varying across trials between 0 ms, ±100 ms, ±200 ms, and ±400 ms.

(all $p < 0.05$; Supplementary Results for Experiment 2; Fig. S6). The IAF correlations with the corresponding visuotactile simultaneity measures were mostly nonsignificant, with the exception of a significant correlation between resting state IAF in the right hemisphere and simultaneity sensitivity.

### Experiment 3: alpha tACS modulates the TBW
In Experiment 3, we tested the causal role of alpha frequencies in shaping body ownership and simultaneity perception. We used tACS to modulate parietal oscillations, either increasing or decreasing alpha frequencies[11,54–57]. The rationale was that if alpha frequency modulates temporal multisensory integration for body ownership and simultaneity in the posterior parietal cortex, then increasing or decreasing alpha oscillations over this region should, respectively, narrow or widen the TBW in the same participants.

Participants underwent three sessions on three different days, separated by at least 24 h. In each session, they received continuous tACS over the posterior parietal cortex at one of three possible frequencies: low alpha (8 Hz), high alpha (13 Hz), or a control non-stimulation condition (sham), while performing body ownership and simultaneity judgment tasks in separate blocks (Fig. 5a, b). EEG Experiment 2 had revealed that the IAF ranged between 9.17 and 12.33 Hz during the perceptual judgment tasks after controlling for the aperiodic component. We therefore selected 8 Hz and 13 Hz frequencies to ensure that, across participants, stimulation frequencies would consistently fall outside their endogenous alpha range. This allowed us to exogenously shift the frequency of ongoing alpha oscillations toward slower or faster rhythms via entrainment, thereby modulating the temporal resolution of perceptual processing during the tasks.

Brain stimulation modulated body ownership ($F_{2,58} = 50.940$; $p < 0.001$; $\eta_p^2 = 0.637$) and simultaneity ($F_{2,58} = 64,726$; $p < 0.001$; $\eta_p^2 = 0.691$) TBWs as hypothesized. Compared with the sham group, the low-alpha-tACS group presented wider body ownership (208,256 vs. 164,981 ms; $p < 0.001$; $t = 5.578$; Fig. 6a, b) and simultaneity (156,305 vs. 125,315; $p < 0.001$; $t = 6.855$; Fig. 6c, d) TBWs. In contrast, high-alpha tACS narrowed body ownership (137,929 ms; $p < 0.001$; $t = 6.673$) and simultaneity (101,415 ms; $p < 0.001$; $t = 6.824$) TBWs. Moreover, participants whose simultaneity TBW was most affected by tACS were also those whose body ownership TBW was most influenced by brain stimulation, further supporting the hypothesis that alpha frequency plays a fundamental role in the temporal resolution of multisensory processing in both tasks (Supplementary Results for Experiment 3; Fig. S7a, b).

### Experiment 3: alpha tACS modulates sensitivity
As expected, tACS modulated both body ownership ($F_{2,58} = 33.562$; $p < 0.001$; $\eta_p^2 = 0.536$; Fig. 7a) and simultaneity ($F_{2,58} = 65.592$; $p < 0.001$; $\eta_p^2 = 0.693$; Fig. 7b) sensitivities. Compared with the sham treatment, low-alpha tACS was associated with reduced body ownership (1.384 vs. 1.676; $p < 0.001$; $t = -4.475$) and simultaneity (2.157 vs. 1.858; $p < 0.001$; $t = -5.249$) sensitivities, which is consistent with the widening of the TBWs. Similarly, high-alpha tACS increased both body ownership sensitivity (1.918; $t = -3.706$; $p < 0.001$) and simultaneity sensitivity (2.509; $p < 0.001$; $t = -6.192$), which is consistent with narrower TBWs. There was no significant tACS effect on bias; As in previous experiments, body ownership and simultaneity TBWs were correlated, as were their sensitivity indices (Supplementary Results for Experiment 3; Fig. S8). Moreover, body ownership was again associated with larger TBW and lower sensitivity than simultaneity. (Supplementary Results for Experiment 3).

### Experiments 2 and 3: Bayesian causal inference model
Building on Experiments 2 and 3, we applied a probabilistic computational model based on Bayesian causal inference to quantify how individual differences in IAF influence perceptual judgments of ownership and simultaneity, offering a principled computational framework to explain the observed behavioral patterns. A prominent view posits that multisensory perception arises from a probabilistic process in which the brain infers a common cause for different unisensory inputs (causal inference)[58–61]. This inference integrates sensory evidence by weighing sensory uncertainty and prior information. In body

## Body ownership judgments

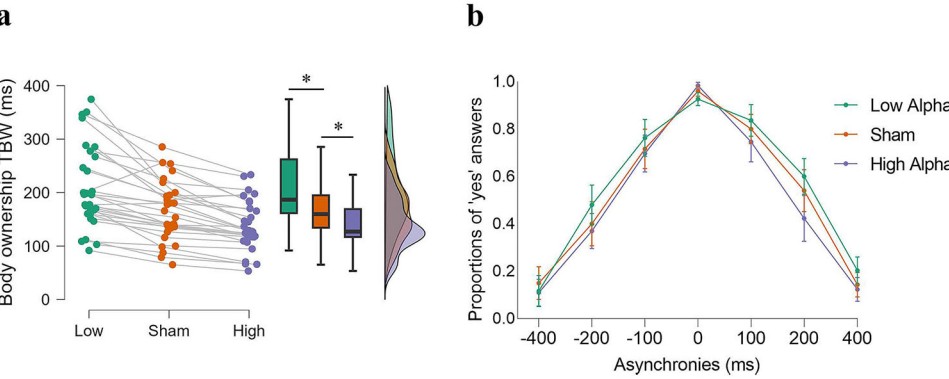

## Simultaneity judgments

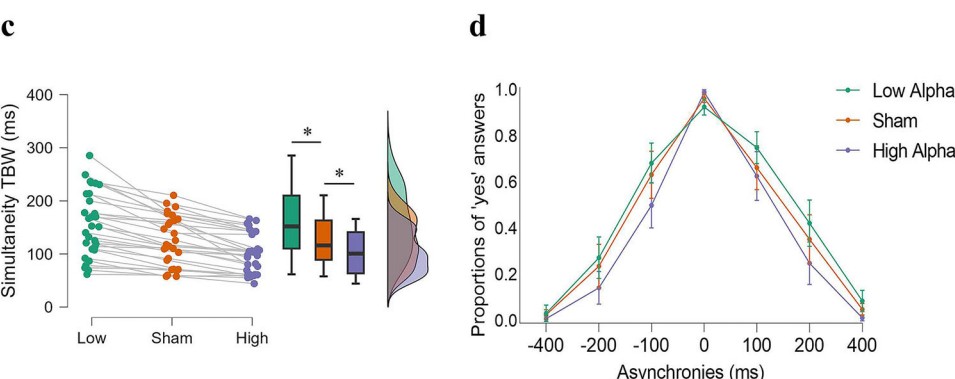

**Fig. 6 | Transcranial alternating current stimulation (tACS) modulation of temporal binding windows (TBWs) in Experiment 3. a, c** tACS modulation of body ownership TBW and visuotactile simultaneity TBW. Asterisks indicate significant differences between stimulation conditions. The central line in each box plot represents the median value. The whiskers represent the most extreme values within 1.5 × IQR from the lower and upper quartiles. A three-way ANOVA revealed a main effect of tACS conditions in both body ownership ($F_{2,58} = 50.940$; $p < 0.001$; $\eta_p^2 = 0.637$; $N = 30$) and visuotactile simultaneity ($F_{2,58} = 64,726$; $p < 0.001$; $\eta_p^2 = 0.691$; $N = 30$). Post-hoc comparisons (Holm-Bonferroni corrected) showed

that, compared to sham, low alpha tACS enlarged the TBWs for both body ownership ($p < 0.001$; $t = 5.578$) and simultaneity ($p < 0.001$; $t = 6.855$) judgments. In contrast, high alpha tACS narrowed the TBW compared to sham in both body ownership ($p < 0.001$; $t = 6.673$) and simultaneity judgments ($p < 0.001$; $t = 6.824$). **b, d** Mean proportions of "yes" answers as a function of the level of asynchrony and tACS stimulation conditions in body ownership judgment tasks and simultaneity judgment tasks. In both graphs, error bars represent the 95% confidence intervals ($N = 30$). Source data are provided as a Source Data file.

ownership, the brain infers a common cause for visual and somatosensory inputs when the rubber hand feels like one's own[29,40,62,63]. We hypothesized that if IAF determines the TBW by acting as a perceptual frame for temporal integration, then a higher IAF would indicate lower uncertainty in temporal visuotactile inputs. To test this hypothesis, we fitted a Bayesian causal inference model that accounts for body ownership and simultaneity judgment tasks[29] to the behavioral data from Experiment 2 (Fig. 8a, b). The model includes parameters for the a priori probability of visuotactile inputs originating from a common source (specific to each type of judgment), one sensory uncertainty parameter (as the same asynchronies were tested for both tasks) and the probability of the observer lapsing (random guessing, specific to each participant). Crucially, we examined whether IAF is related to these model parameters estimated at the individual level, focusing on whether IAF and sensory uncertainty exhibit the expected relationship.

We found that sensory uncertainty (σ) correlated with parietal IAF measured during body ownership judgment tasks in the left hemisphere (Pearson's $r = -0.453$; $p = 0.002$, Spearman' $\rho$:-453; $p = 0.002$; Fig. 8c) and right hemisphere (Pearson's $r = -0.339$; $p = 0.029$; Spearman's $\rho$:-0.322; $p = 0.029$). Parietal IAF measured during simultaneity judgment tasks also correlated with sensory uncertainty (σ) in the left hemisphere (Pearson's $r = -0.400$; $p = 0.006$; Spearman's $\rho = -0.402$;

$p = 0.006$; Fig. 8d) and right hemisphere (Pearson's $r = -0.372$; $p = 0.011$; Spearman's $\rho = -0.385$; $p = 0.008$). Resting-state IAF correlated with sensory uncertainty (σ), although this correlation reached significance only in the left parietal cortex (Pearson's $r = -0.437$; $p = 0.002$; Spearman's $\rho = -0.489$; $p = 0.001$). FOOOF-IAF analyses provided convergent results (Supplementary Results for Experiment 2; Fig. S5). There was no significant correlation between IAF (under resting-state or task conditions) and the a priori probability that vision and touch come from a common source ($p_{same}$) nor between sensory uncertainty and alpha power (Supplementary Results for Experiment 2). Thus, IAF is specifically related to the uncertainty carried in the asynchronous visuotactile signal (σ). This result aligns well with the findings on how IAF is related to TBWs and sensitivities, while offering a deeper computational understanding of how alpha oscillations may constrain the temporal resolution of sensory integration via their relationship with internal sensory uncertainty. Next, we examined the effects of tACS on the causal inference model's parameters. Our goal was to determine whether the behavioral response observed when altering alpha frequency through tACS—the widening or narrowing of TBWs—was better explained by changes in sensory uncertainty (σ) or changes in the prior probability of a common cause for visuotactile inputs. In line with our hypothesis, we found that the changes in TBWs

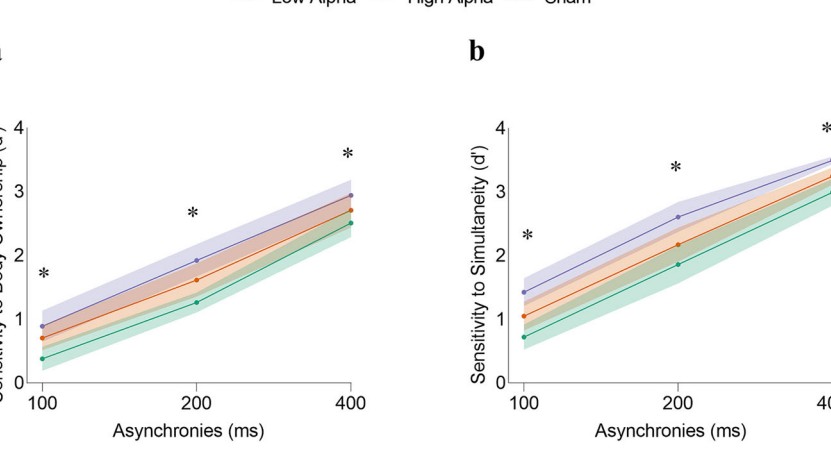

**Fig. 7 | Transcranial Alternating Current Stimulation (tACS) modulation of perceptual sensitivity in Experiment 3. a**, **b** Mean sensitivity as a function of asynchronies (100 ms; 200 ms; 400 ms) and the three stimulation conditions (green: low alpha; purple: high alpha; orange sham). Shaded regions represent the 95% confidence intervals. Asterisks indicate significant differences between the three stimulation conditions. Two separate ANOVAs for body ownership sensitivity and visuotactile simultaneity sensitivity, with stimulation and asynchrony as factors, revealed that tACS modulated both body ownership ($F_{2,58} = 33.562$;

$p < 0.001$; $\eta_p^2 = 0.536$; $N = 30$) and simultaneity ($F_{2,58} = 65.592$; $p < 0.001$; $\eta_p^2 = 0.693$; $N = 30$) sensitivities. Post-hoc comparisons (Holm-Bonferroni corrected) showed that, compared with sham, low alpha tACS reduced body ownership ($p < 0.001$; $t = -4.475$) and simultaneity ($p < 0.001$; $t = -5.249$) sensitivities, whereas high-alpha tACS increased body ownership sensitivity ($t = -3.706$; $p < 0.001$) and simultaneity sensitivity ($p < 0.001$; $t = -6.192$). Source data are provided as a Source Data file.

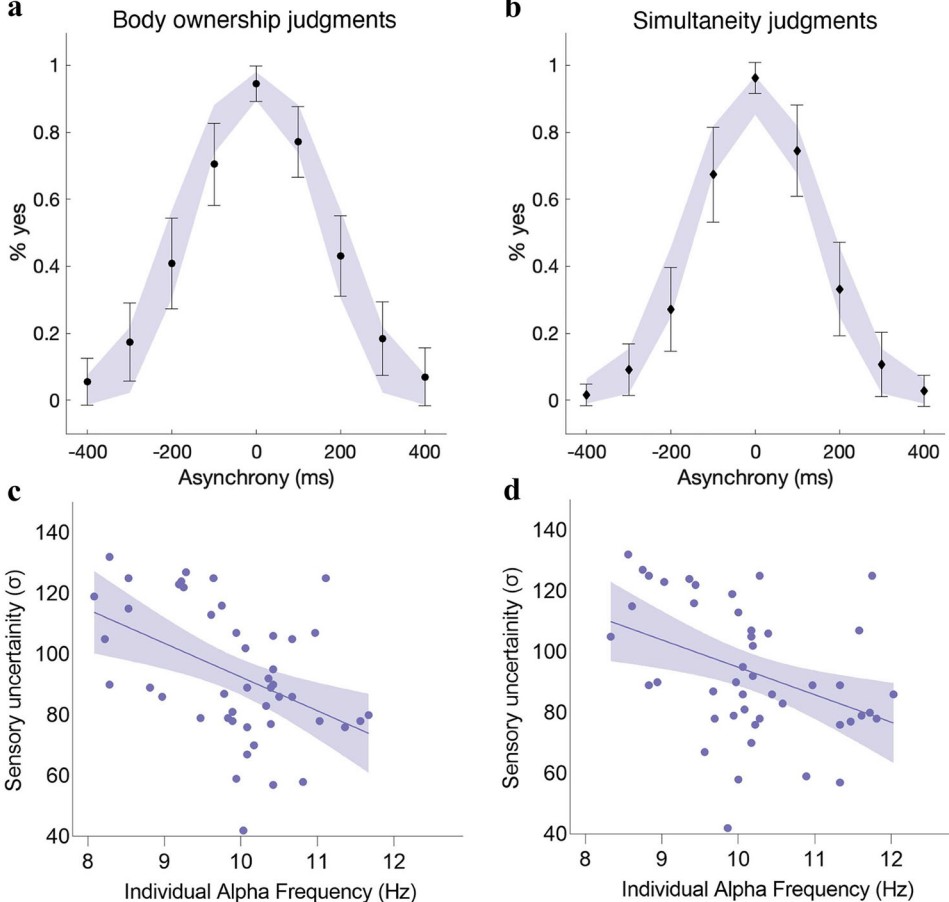

**Fig. 8 | Results of the computational modeling in Experiment 2. a**, **b** Observed and predicted responses ($N = 46$) for body ownership and simultaneity judgment tasks. Black dots (**a**) and diamonds (**b**) represent the mean of the observed data. In both graphs, the error bars represent the standard deviation (+/-SDs). The shaded areas represent the corresponding Bayesian causal inference model predictions (+/-SDs). **c**, **d** Correlation between sensory uncertainty (σ) and individual alpha

frequency measured during body ownership (Pearson's $r = -0.453$; $p = 0.002$; $N = 46$) and simultaneity judgment tasks (Pearson's $r = -0.400$; $p = 0.006$; $N = 46$). In both graphs, the solid line represents the best-fitting regression. The shaded regions reflect the 95% confidence interval. Source data are provided as a Source Data file.

induced by tACS were better explained by a model that assumed changes in the level of sensory uncertainty rather than by a model that assumed changes in the a priori probability for a common source. Both the Akaike information criterion (AIC[64]) and the Bayesian information criterion (BIC[65]) were lower for the model with varying uncertainty levels than for the model with different priors, indicating that the former outperformed the latter when correcting for the number of free parameters (Fig. S2, Table S1). These results suggest that tACS influences the uncertainty of temporal visuotactile input in causal inference processes, in line with the direction of alpha frequency manipulation—speeding up or slowing down alpha frequency—within the Bayesian causal inference framework. This finding reinforces the notion that IAF modulates body ownership and simultaneity by affecting the temporal resolution of multisensory processing.

## Discussion

The current psychophysical, EEG, tACS, and computational model results provide conclusive evidence that alpha frequencies play crucial roles in the temporal integration of multisensory information for body ownership and visuotactile simultaneity perception. IAF predicted the TBW and sensitivity in both body ownership and simultaneity judgments, and tACS provided causal evidence that slowing down or speeding up alpha frequencies leads to corresponding changes in TBWs and sensitivities. Computational modeling indicated that the effect of alpha frequencies on visuotactile temporal integration is best captured by a change in sensory uncertainty, with faster frequencies associated with less uncertain visuotactile asynchrony information, and vice versa. These findings are significant, as they show that brain oscillations—parietal alpha frequency specifically—play a crucial role in how we perceive our own bodies and distinguish ourselves from the external world while also contributing to resolving the debate over the role of alpha oscillations in the temporal integration of multisensory signals.

To generate body ownership, the brain addresses the perceptual binding problem by integrating signals originating from one's body while treating signals from external events separately[29]. The current findings advance our understanding of the basic electrophysiological mechanism involved in solving this binding problem by showing that parietal alpha frequency determines the temporal integration of somatosensory and self-related visual information, thereby influencing the multisensory perception of a limb as one's own. In line with the literature on IAF in visual and audiovisual perception[11–17], our findings suggest that a higher frequency translates to a higher temporal resolution, as more multisensory information can be sampled over the same time period through more perceptual cycles. In our experiments, visuotactile asynchrony is perceived differently depending on the number of perceptual sweeps over the cross-modal stimulus period, which are determined by IAF. An individual with a 9 Hz IAF has a sampling period of ~110 ms, resulting in fewer perceptual frames during the relevant stimuli presentation than an individual with a 90 ms sampling period (~11 Hz) and consequently less perceptual evidence to infer whether the visuotactile information originates from different or the same source. Our EEG, tACS, and psychophysical results support this view, linking IAF to TBWs and perceptual sensitivity indices in both simultaneity and body ownership judgment tasks. Although the judgments in the latter task do not pertain to simultaneity but rather to illusory hand ownership, they are nevertheless driven by asynchrony information in the visuotactile inputs.

Our computational modeling results provide insight into how IAF may shape the temporal integration of multisensory signals by modulating sensory uncertainty involved in perceptual inference. According to the Bayesian causal inference model of perception[58–61], the brain solves the multisensory binding problem by inferring the causes of its sensory input based on the available sensory evidence and prior beliefs[58,60]. Sensory uncertainty plays a crucial role in this probabilistic

process, as it determines the inferred probability of a common cause. Previous computational psychophysics experiments have demonstrated that the brain uses this probabilistic approach in causal inference of body ownership[29,39,40]; illusory hand ownership is perceived when there is a high inferred probability of a common cause for somatosensory and visual inputs. Crucially, we found that IAF was related to sensory uncertainty when we fitted a causal inference model[29] to the behavioral data. Specifically, we found that a higher IAF was associated with less uncertain visuotactile asynchrony information and, conversely, that a lower frequency was related to greater uncertainty. This observation fits well with the links between IAF, TBWs and the perceptual sensitivities we observed and, importantly, provides a deeper understanding for how alpha oscillations—a physiological process—are related to the computational processes whereby the brain solves the multisensory temporal binding problem. The tACS results were consistent with this conclusion, suggesting that the effects of brain stimulation speeding up or slowing down IAF could best be captured as changes in sensory uncertainty in the causal inference. Our finding that IAF is linked to uncertainty in multisensory asynchrony information extends our understanding of how brain oscillations relate to the 'Bayesian Brain'[58–61] and 'Predictive Coding'[66,67] hypotheses. While previous studies have suggested that alpha power is associated with top-down predictions and prior beliefs[68], the role of IAF in sensory uncertainty has only been theorized[69]; the current findings provide valuable experimental support for this connection.

Our findings offer a new way of conceptualizing individual differences in multisensory perception and how these differences relate to brain oscillations. Individuals with a faster IAF are likely to display a more "sensory-driven" perceptual inference in the temporal domain, as asynchrony information is less uncertain and therefore perception is less influenced by priors, whereas individuals with a slower IAF are likely to show the opposite pattern, with perception being relatively more influenced by priors due to more uncertain multisensory temporal signals. This insight may be relevant for understanding disturbances in multisensory perception in clinical populations, such as individuals with schizophrenia. A slower IAF in posterior brain regions has been reported in the schizophrenic population[70,71], which suggests that the multisensory temporal asynchrony signals of individuals with schizophrenia are more uncertain. This increased uncertainty may drive "overreliance" on prior knowledge, potentially contributing to aberrant perceptual experiences in this group[72,73].

Both the EEG analysis and tACS focused on the posterior parietal cortex; thus, our results advance the understanding of this region's mechanistic role in body ownership perception and multisensory integration. A previous EEG study revealed alpha power modulation during the rubber hand illusion[53]. However, alpha desynchronization was interpreted as multisensory area activation, leaving the mechanism through which the parietal cortex integrates self-related sensory signals unclear. Previous fMRI[34–39] and ECoG[42] experiments have highlighted activity in the posterior parietal cortex that reflects the temporal integration of visuotactile signals in the rubber hand illusion, but the neurophysiological mechanism by which this activity is achieved remains poorly understood. Our results indicate a mechanistic role of parietal alpha frequency in temporally integrating visuotactile signals for body ownership and simultaneity perception. Alpha oscillations can modulate local neuronal excitability and spiking activity[74,75], thereby influencing how parietal neurons integrate visuotactile signals. Parietal alpha waves may also allow task-relevant information flow between brain regions through cross-frequency interactions[76,77]. Regardless of the exact underlying physiological mechanism facilitating local integration or neural communication, our findings indicate an important role of parietal alpha oscillations in the temporal integration of bodily sensory signals.

However, we do not claim that the PPC is the only cortical region where IAF may influence visuotactile integration. Similar mechanisms

may operate in other multisensory areas, such as the ventral premotor cortex, which has been particularly implicated in the feeling of body ownership[34,40]. An exploratory post-hoc analysis revealed that IAF over the premotor cortex showed significant correlations with body ownership TBWs and sensitivities (Supplementary Results for Experiment 2; Fig. S6), but not with the corresponding visuotactile simultaneity measures (Supplementary Results for Experiment 2). This finding suggests that the role of IAF in shaping temporal perception extends beyond the PPC, at least for body ownership. The apparent selectivity of premotor IAF for body ownership aligns well with previous literature showing that activity in this area is related to body ownership beyond visuotactile synchrony/asynchrony effects[34,37]; for instance, Guterstam and colleagues, using ECoG[42], found that RHI-related PPC activity is closely linked to the individual visuotactile stimulations delivered to participants' real and rubber hands, whereas premotor activity is more sustained between individual touches and reflects the continuous feeling of hand ownership[42]

Our findings provide important insights into the debate regarding the association between IAF and temporal integration in perception, advancing our understanding of this relationship. A recent study found no evidence that IAF was related to perceptual sensitivity in either visual or cross-modal contexts[21], which sparked discussions in the field[22-27]. Although we did not investigate temporal processing within a single sensory modality, our findings provide strong evidence supporting the hypothesis that IAF modulates the temporal integration of multisensory inputs. Previous studies reporting significant associations between IAF and TBW[10-20,55,56] have been criticized[21,26] for relying on heterogeneous behavioral and psychophysical methods, not ruling out cognitive bias, using different methods to estimate IAF, and being vague about the mechanism by which IAF influences perception. Our study addresses these critiques by presenting converging evidence from several complementary methods. First, perceptual performance was analyzed using psychophysical indices of TBW and bias-free perceptual sensitivity. Second, computational modeling supported our conclusions and provided a formal account of how IAF modulates perception through its relationship with sensory uncertainty. Third, IAF was estimated using two methods: we defined IAF as the frequency with the maximum power within the alpha range, and we employed the FOOOF algorithm to isolate the oscillatory components from aperiodic background activity[50]. Finally, we used tACS to demonstrate a causal relationship, further strengthening the evidence for a mechanistic link between IAF and the temporal resolution of multisensory perception. Regarding the tACS experiment, it is important to highlight the substantial inter-individual variability in stimulation effects, ranging from no observable effect to marked behavioral changes. This variability may reflect differences in how stimulation at the edges of the alpha band (8 Hz and 13 Hz) interacted with each participant's intrinsic alpha peak. It is therefore possible that targeting frequencies closer to the IAF (e.g., ±1−2 Hz) would have yielded stronger behavioral effects[11,14,55]. Nevertheless, stimulation at 8 Hz and 13 Hz was still effective in producing significant group-level modulation, consistent with previous tACS studies that applied stimulation near the boundaries of the target frequency band[54,57,78]. For example, Venskus et al. (2022)[57] applied tACS at low (8 Hz) and high (14 Hz) alpha frequencies and observed significant modulation of the temporal binding window associated with the sense of agency. Similarly, Wolinski et al. (2018)[78] applied theta-band stimulation (4 Hz and 7 Hz) and reported corresponding changes in visuospatial working memory capacity.

In summary, our study provides converging psychophysical, correlational (EEG), and causal (tACS) evidence that parietal IAF modulates the temporal integration of visual and somatosensory information in both body ownership and simultaneity perception. Our computational modeling results suggest that this modulation arises from the influence of alpha frequency on the uncertainty of multisensory asynchrony information that determines the integration

process. These results establish that alpha frequency plays a pivotal role in bodily self-awareness by influencing the temporal integration versus segregation of bodily signals—crucial for perceiving one's body as one's own and distinguishing between external and self-related sensory information. Brain oscillations, therefore, are fundamental to how the brain constructs a coherent sense of bodily self.

## Methods
### Experiment 1
**Participants.** Thirty-four participants were recruited for the experiment (19 females; ages 21−37 y; mean age 28.266 y). However, since the ability to experience the rubber hand illusion was required to perform the task and to fit the psychometric functions to the response data, we included in the main experiment only participants (N = 30; 17 females; ages 21−37 y; mean age 28.26 y) that fulfilled the inclusion criterion of being able to experience the rubber hand illusion (participants' inclusion test in the Supplementary Methods). All participants had no history of neurological or psychiatric disorders and provided written informed consent to take part in our experiment, which was approved by the Swedish Ethical Review Authority. Participants received monetary compensation to complete the experiment. A power analysis was performed for sample size estimation using G*Power, which revealed a correlation with an effect size of 0.5 and a power level of 0.80[79]. The analysis indicated a critical sample size of 29 subjects, which was increased to 30 to identify a correlation between tasks with a power >0.80.

**Experimental setup and procedure.** Participants engaged in two different tasks: *(i)* a body ownership judgment task and *(ii)* a simultaneity judgment task. The order of the tasks was counterbalanced across participants. In both tasks, participants maintained a similar position, with their right hand laying palm down on a flat support 30 cm lateral to the body midline. A chin rest and an elbow rest ensured that the participants' head and arm remained in a steady and relaxed position. During body ownership judgment tasks, a rubber right hand—a cosmetic prosthetic right hand (model 30916-R, Fillauer®) filled with plaster—was placed in view on a platform 15 cm directly above and aligned with the real hand that was placed on a lower platform out of view. Two robotic arms applied tactile stimuli (taps) to the rubber hand's index finger and to the participant's hidden real index finger. Each robotic arm was composed of three parts: two 17-cm-long, 3-cm-wide metal pieces and a metal slab (10.20 cm) as a support. The joint between the two metal pieces and that between the proximal piece and the support were powered by two HS-7950TH Ultra Torque servos that included 7.4 V optimized coreless motors (Hitec Multiplex, USA). The distal metal piece ended with a ring containing a plastic tube (diameter: 7 mm) that was used to precisely touch the rubber hand and the participant's real hand. Taps on the rubber hand were either synchronized with the taps on the participant's real hand (synchronous condition) or were delayed or advanced at different asynchronies. The asynchronies varied across trials in five steps between ±500, ±300, ±150, ±50, and 0 (true synchrony) ms, where a negative value indicates that the rubber hand was touched first, and a positive value indicates that the participant's hand was touched first.

In each trial, the robotic arms tapped the rubber hand's index finger and the participant's index finger six times each for a total period of 12 s (i.e., at a frequency of 0.5 Hz) at corresponding locations. The robotic arms tapped the corresponding parts of the real and rubber fingers, targeting four different locations in randomized order (immediately proximal to the nail on the distal phalanx, on the middle phalanx, on the proximal interphalangeal joint, and on the proximal phalanx) to avoid skin irritation. The participant was instructed to focus their gaze on the rubber hand. Then, the robots stopped while the participant heard a tone instructing them to verbally report whether the rubber hand felt like their own hand by saying "yes" ("the

rubber hand felt like it was my hand") or "no" ("the rubber hand did not feel like it was my hand"). After the stimulation period and the body ownership judgment answer, the participant was asked to wiggle their right fingers to break the illusion, eliminating possible carry-over effects and reducing the risk of muscle stiffness. Participants were also asked to relax their gaze and look away from the rubber hand if they wanted. Five seconds later, a second tone informed the participant that the next trial was about to start, and the next trial started 1 s after this sound cue. The trials were divided into three blocks consisting of 4 repetitions for each asynchrony; thus, 36 trials were acquired per block.

In the simultaneity judgment task, the participant's right hand lay on a flat support, palm down. The participant judged the perceived simultaneity of two paired stimuli, i.e., a visual stimulus and a tactile stimulus. Tactile stimuli were delivered on the right index fingertip through a small vibrating motor with a 10 mm diameter (Precision Microdrives vibration motor, model 310−113, 3.6 V, 70 mA, 240 Hz), placed on a small foam support. Visual stimuli consisted of a red-light-emitting diode placed over a similar foam support, 2 cm from the tip of the participants' index finger. Each trial began with a brief presentation of either a visual or tactile stimulus lasting 30 ms, followed after variable asynchrony by a 30-ms stimulus in the other sensory modality. Immediately after stimuli presentation, participants were instructed to verbally report whether the two stimuli were synchronous by saying "yes" ("the visual and tactile stimuli were synchronized") or "no" ("the visual and tactile stimuli were not synchronized"). The next trial began after a variable interval randomly ranging from 2000−3500 ms to prevent participants from anticipating the stimulus presentations. The asynchronies used in the simultaneity judgment task were in the same temporal range as the rubber hand illusion, i.e., from perfect synchronicity (0 ms) up to 500 ms. However, the intervals were adjusted to better capture the simultaneity task, which requires additional brief delays to better fit the psychometric functions[29]. Thus, the level of asynchrony in this task varied across trials between ±500, ±400, ±300, ±200, ±150, ±100, ±50, and 0 (true synchrony) ms. Trials were divided into two blocks, with 12 repetitions for each asynchrony, resulting in a total of 180 trials per block.

**Data Analysis.** In both tasks, the TBW was obtained as the standard deviation of the Gaussian curve fitted to the proportions of "yes" responses as a function of the asynchronies[3–6]. Therefore, in the simultaneity judgment tasks, the TBW represents the standard deviation of the Gaussian curve fitted to participants' reports of synchrony. In the body ownership judgment tasks, the TBW was the standard deviation of the Gaussian curve fitted to the proportions of trials in which participants reported feeling the rubber hand illusion. The Gaussian curve showed an excellent fit to the data in both body ownership (mean $R^2 = 0.953$; range = 0.850−0.989) and simultaneity judgments (mean $R^2 = 0.962$; range = 0.920−0.985). We ascertained the presence of a correlation between the two TBWs through Pearson and Spearman correlation analyses. Additionally, we compared body ownership and simultaneity TBWs using a paired sample $t$-test. Based on a previous study[29], we expected the TBW to be wider for body ownership, as the RHI also involves contextual sensory information and perceptual priors regarding the relative orientation and placements of the rubber hand real hands (visuoproprioceptive spatial cues)[61,80].

In addition to the TBW, we computed d' scores for each of the asynchronous conditions, collapsing positive and negative asynchronies ( ≠ 0 ms; noise) with respect to the synchronous condition ( = 0 ms; signal). Here, d' was calculated as $d' = z(H) − z(FA)$, where $z(H)$ represents the z score of the hit rate, i.e., the probability of correct reactions (i.e., *the rubber hand felt like my own hand* in the body ownership judgment task or *the touches were synchronous* in the synchronicity judgment task) in 0-ms trials, and the $z(FA)$ represents the z

score of false alarms in asynchronous trials[31]. To avoid zero counts, we applied padding (edge correction) by either adding or subtracting half a trial[81]. Finally, we computed the area under the curve of the d' values as a function of the asynchronies (AUC d') to provide an index of the sensitivity to the simultaneity and body ownership judgment tasks. We ascertained a correlation by computing Pearson and Spearman correlations. Additionally, we compared sensitivity to body ownership and sensitivity to simultaneity using a paired sample $t$-test. Consistently with the TBW, we expected that AUC d' for body ownership to be lower than that for visutacitle simultaneity. Moreover, we checked whether the sensitivities to body ownership and simultaneity judgment tasks increased with increasing asynchronies−as expected from previous studies[31]−with two separate ANOVAs on the d' scores with asynchronies as a factor.

### Experiment 2

**Participants.** We recruited another group of fifty-seven participants (31 women; ages 18–43 y; mean age 27.701 y). As in Experiment 1, only participants ($N = 46$; 26 women; ages 18–43 y; mean age 27.978 y) who fulfilled the inclusion criterion of being able to experience the rubber hand illusion were included in the main experiment. Similar to Experiment 1, the power analysis indicated a sample size of 29 participants, indicating a correlation with an effect size of 0.5 and a power level of 0.80. However, we increased the sample size to 46 to achieve a statistical power >0.80 for identifying correlations between tasks. Additionally, this larger sample size accounted for the expectation that the FOOOF analysis might fail to detect a clear task-related alpha peak in 10–20% of the participants. All participants had no history of neurological or psychiatric disorders and provided written informed consent to participate in our behavioral study. The experiment was approved by the Swedish Ethical Review Authority. Participants received monetary compensation to complete the experiment

**Experimental setup and procedure.** Before starting Experiment 2, we recorded 4 min of resting-state EEG data. Participants were instructed to relax with their eyes open, focusing on a fixation cross positioned on the wall ~75 cm away. After that, participants performed body ownership and simultaneity judgment tasks. The experimental setup was identical to that of Experiment 1, including the placements of the rubber hand and real hand and the use of robotic arms. One adjustment to the procedure was that in both tasks, the asynchronies between the rubber hand and the participant hand varied across trials between ±400, ±300, ±200, ±100 and 0 ms. Thus, the visuotactile stimulation used in the simultaneity judgment task was identical to that used for the body ownership judgment task to match the tasks as closely as possible. Additionally, in Experiment 2, the interval between one touch and the other was jittered between 1100 and 1300 ms to avoid cortical oscillation entrainment. Body ownership and simultaneity judgment tasks were administered in two different blocks of trials using the ABAABB counterbalancing order, with the order counterbalanced across participants. Each asynchrony was repeated 5 times in each block, leading to a total of 45 trials per block.

**EEG recording and preprocessing.** Both resting-state and task-related EEG data were recorded and digitized at a sampling rate of 1024 Hz using a 128-electrode Biosemi system with an elastic cap, in which electrodes were integrated at sites conforming to the ABC system. All impedance values were kept below 50 kΩ. Scalp electrodes were referenced to A1. The continuous EEG data were resampled to 512 Hz and then filtered, leaving frequencies between 0.3 and 40 Hz, and epoched from -500 ms before to 500 ms after each touch was applied to the participant's real hand. Electrodes with a variance smaller than -2 or larger than 2 standard deviations of the mean activity

of all electrodes were rejected. To identify EEG artifacts, we implemented an independent-component analysis (ICA[82]) on the epoched data. We evaluated each component using the automated ICLabel classifier[83] and rejected all components that exhibited a 75% or higher probability of being associated with eye blinks, ocular movements, muscle activity, heartbeat, or channel noise. EEG electrodes previously rejected were replaced with the weighted average activity in all remaining electrodes using spherical spline interpolation. On average, 13% [SD = 8] of electrodes were rejected and subsequently interpolated per participant. However, no electrodes were rejected in our two ROIs. Epochs with voltages exceeding ±150 μV were excluded from further analysis; on average, 1.4% [1.6] of trials were excluded per participant. Finally, the EEG signal was rereferenced to the average across all electrodes. Preprocessing was performed using custom-made MATLAB code (R2023b, MathWorks, Inc.) and code developed for the EEGLAB toolbox[84].

**EEG power spectrum density analysis.** We computed the power spectrum density (PSD) for the alpha band (8–13 Hz) using the modified Welch periodogram method implemented in the MATLAB 'pwelch' function and a fast Fourier transform (FFT). To have a nominal frequency of 0.167 Hz, we segmented the preprocessed EEG signal into 6-s windows with zero padding and 50% overlap. Next, we converted the absolute PSD values into relative power by dividing each participant's alpha power by the total power across all frequencies included in the analysis, allowing for direct comparisons between subjects and conditions. To identify IAF and power during the task, we segmented the EEG signal in 1000-ms epochs (from -500 to +500 relative to each touch applied to the participants' hand). Power was normalized by z score decibel transformation. IAF was determined by identifying the peak frequency in the alpha frequency range for each electrode, which corresponds to the frequency band with the highest power value. Based on prior findings that posterior parietal cortices are implicated in multisensory integration underlying body ownership[34–39], we selected electrodes from these regions for further analysis. We defined two regions of interest (ROIs): A5, A6, A7, A8, A17, and A18 in the left hemisphere and B3, B4, B5, A30, A31, and A32 in the right hemisphere. These data were subsequently exported for statistical analysis. PSD and FFT were performed using custom-made MATLAB code and code developed for the Brainstorm toolbox[85].

**Data analysis: assessing correlations of IAF with TBWs and sensitivities.** Our hypothesis posits that alpha frequencies sample visuotactile information across time to shape body ownership and multisensory simultaneity perception; therefore, we expected significant correlations between the TBWs measured in body ownership and simultaneity judgment tasks and IAF identified in both the EEG data acquired under resting-state and task conditions. We ascertained a significant correlation by computing Pearson and Spearman correlations between TBWs and the IAF measured in both hemispheres during perceptual judgment tasks and during the resting-state period. Since we anticipated that these correlations are related to individual differences in multisensory perception rather than bias, we additionally predicted a correlation between IAF and sensitivities to body ownership and visuotactile simultaneity. Therefore, we employed a similar correlational approach for the sensitivity to body ownership and simultaneity judgment tasks (AUC d') and for the bias indices associated with each task. We estimated the bias for each asynchrony with the formula $bias = -0,5 (Z(H) + Z(FA))$. Positive and negative values different from 0 indicate the presence of bias, with negative values indicating a more liberal decision criterion (cognitive bias) or a stronger "baseline" RHI induced by just looking at the rubber hand before the visuotactile stimulation starts (perceptual bias). We averaged the values of bias across the asynchronies to obtain an index of

the bias associated with the simultaneity and body ownership judgment tasks. Moreover, also in Experiment 2 the Gaussian fit showed a good fit to the data with a mean $R^2$ of 0.974 for body ownership (range: 0.867 – 0.997) and 0.976 for simultaneity judgments (range: 0.920 - 0.997).

## Experiment 3

**Participants.** Another group of thirty-seven participants were recruited for the study (20 women; ages 18-40 y; mean age 26.43 y). Only participants who met the inclusion criteria for the rubber hand illusion were included in the main experiment (N = 30; 16 women; ages 18-39 y; mean age 25.53 y). None of the participants met any of the exclusion criteria for tACS[86–89], and all provided written informed consent to take part in our study, which was approved by the Swedish Ethics Review Authority. None of the participants had a past history of neurological or psychiatric disorders or were under regular medication as established by a self-report questionnaire prior to participating in the tACS experiment. Participants received monetary compensation to complete the experiment. Additionally, in this case, the sample size was determined with a power analysis using G*Power, which specified a medium effect size of 0.25 and a power level of 0.80 for 3 measurements (low alpha, high alpha, and sham). The analysis indicated that a sample size of 28 was needed. We increased the sample size to 30 to allow for a 10% anticipated dropout rate and to identify the effects of tACS with a power >0.80.

**Experimental setup and procedure.** In the tACS experiment, the visuotactile stimulation for body ownership and simultaneity judgment tasks was identical to that used in Experiment 2. However, the asynchronies between participants and the rubber hand varied across trials between ±400, ±200, ±100 and 0 ms. The intertrial interval was reduced to 4 s, and the interval between one touch and the other randomly varied between 1000 and 1200 ms. These small changes were introduced to reduce the duration of the experimental session with brain stimulation, ensuring that it did not exceed 40 min. In each session, body ownership and simultaneity judgment tasks were administered in two separate blocks of trials using the ABBA counterbalancing order, with the order counterbalanced across participants. Each asynchrony was repeated 5 times in each block, resulting in a total of 35 trials per block.

**Transcranial Alternating Current Stimulation (tACS).** tACS was delivered by a battery-powered DC stimulator (Version DC-Stimulator Plus, Neuroconn, Ilmenau, Germany) through a pair of rubber electrodes (5 × 7 cm, 35 cm²) enclosed in saline-soaked sponges and fixed on the head by elastic bands. Based on the standard 10–20 EEG procedure, the electrodes were positioned in the target areas corresponding to posterior parietal regions, specifically over P3 and P4, which overlap with the two ROIs analyzed in the EEG experiment named previously with the ABC EEG layout. For low and high alpha frequencies, participants were stimulated with an in-phase (0°) alternate current mode for 40 min for each frequency, with a fade in/out period of 20 s and a current strength of 1000 μA. Under the sham conditions, the stimulation consisted only of the fade in/out period (i.e., 40 s total), but participants maintained the same setting as the conditions in which they received the real stimulation. The order of the stimulation was pseudorandomized across participants. During the entire time course of the study, participants were not told whether they received real or sham stimulation. None of the participants were able to discriminate between the sham and active stimulation conditions, based on a validated questionnaire that assesses participants' sensation during the stimulation period[87,89] (Supplementary Results for Experiment 3). Moreover, no participants reported any other sensations related to brain stimulation during both low- and high-alpha tACS.

**Data Analysis.** The data analyses closely followed the procedures of previous experiments. For each participant, we computed the width of the TBW in both body ownership and simultaneity judgment tasks across the three stimulation conditions (low alpha, sham, and high alpha). Based on the Experiment 2 EEG results, we expected low-alpha tACS to widen the TBW in both tasks and high-alpha tACS to narrow the TBW. To test this hypothesis, we conducted two separate ANOVAs on the TBWs of body ownership and visuotactile simultaneity judgment tasks with stimulation (low alpha, high alpha and sham) as a factor. Holm-Bonferroni correction was applied to investigate any significant differences. Like in previous experiments, we performed statistical analysis of the d' scores. We conducted two separate ANOVAs for body ownership sensitivity and visuotactile simultaneity sensitivity, with stimulation (low alpha, high alpha, and sham) and asynchrony ($\pm 100$, $\pm 200$, and $\pm 400$ ms) as factors. Moreover, for the sake of completeness, we conducted two repeated-measure ANOVAs on the bias in body ownership and visuotactile simultaneity, using stimulation and asynchrony as factors, to investigate whether tACS affects bias, even though we had no reason to expect it would.

### Experiments 2 and 3: Bayesian causal inference model

Chancel, Ma and Ehrsson[29] developed a Bayesian causal inference model that successfully describes how the perceptual system infers a common cause of visual and tactile signals originating from one's hand and how this relates to the subjective feeling of hand ownership in the rubber hand illusion and the perception of simultaneity between the tactile and visual inputs. We used the same core Bayesian causal inference (BCI) model and the same procedure to fit this model to the participants' responses in the present study. As briefly outlined in the results section, this model included parameters for sensory uncertainty (one parameter for both tasks since the same levels of asynchrony were tested for both tasks), a priori probability of visual and tactile inputs originating from a common source (one for each type of judgment), and the probability of the observer lapsing (randomly guessing; one parameter for both tasks since this parameter is specific to each participant). Here, the aim was to investigate (1) the relationship between the extracted IAF and the parameter values estimated for each participant (using the data from Experiment 2) and (2) which parameters were impacted by tACS stimulation (using the data from Experiment 3). Below, we briefly describe our modeling approach; more details can be found in Chancel et al.[29].

Bayesian inference is based on a generative model, i.e., a statistical model of the world that the observer believes gives rise to observations. By "inverting" this model for a given set of observations, the observer can make an "educated guess" about a hidden state. In our case, the model contained three variables: the causal structure category ($C$), the tested asynchrony ($s$), and the measurement of this asynchrony ($x$) by the participant. The a priori probability of a common cause for vision and touch ($C = 1$) was a free parameter denoted $p_{\text{same\_O}}$ and $p_{\text{same\_S}}$ for the body ownership and simultaneity tasks, respectively. Next, we assumed that for the observer, when $C = 1$, the asynchrony $s$ was always 0. When $C = 2$, we assumed that asynchrony was normally distributed with the correct standard deviation $\sigma_S$ (i.e., the true standard deviation of the stimuli used in this experiment, 293 ms in Experiment 2 and 290 ms in Experiment 3). In other words, $p(s|C = 2) = N(s; 0, \sigma_S^2)$. Next, we assumed that the observer made a noisy measurement $x$ of the asynchrony. We made the standard assumption (inspired by the central limit theorem) that this noise adhered to the following normal distribution:

$$p(x|s) = N(x; s, \sigma^2) \tag{1}$$

where the variance depended on the sensory noise for a given trial. Since the same asynchronies were tested for the body ownership and

simultaneity judgment tasks, we assumed the variance to be identical for both tasks.

From this generative model, we turned to inference. Visual and tactile inputs were integrated, leading to the emergence of the rubber hand illusion if the observer inferred a common cause ($C = 1$) for both sensory inputs. On a given trial and for a given task, the model observer used $x$ to infer the category $C$. Specifically, the model observer computed the posterior probabilities of both categories, $p(C = 1|x)$ and $p(C = 2|x)$, i.e., the belief that the category was $C$. Then, the observer would report $C = 1$, i.e., "yes, it felt like the rubber hand was my own hand" or "yes, the seen and felt touches were synchronous" for the body ownership and simultaneity judgment tasks, respectively, if the former probability was higher, or in other words, if $d > 0$, where

$$d = \log \frac{p(C = 1|x)}{p(C = 2|x)} \tag{2}$$

The decision rule $d > 0$ was thus equivalent to

$$|x| < \sqrt{K} \tag{3}$$

where

$$K = \frac{\sigma^2(\sigma_S^2 + \sigma^2)}{\sigma_S^2} \left( 2 \log \frac{p_{\text{same}_O}}{1 - p_{\text{same}_S}} + \log \frac{\sigma_S^2 + \sigma^2}{\sigma^2} \right) \tag{4}$$

where $\sigma$ was the sensory noise level of the trial under consideration. Consequently, the decision criterion changed as a function of the sensory noise affecting the observer's measurement and the prior probability for a common cause associated with the requested judgment. The output of the BCI model was the probability of the observer reporting the visual and tactile inputs as emerging from the same source when presented with a specific asynchrony value $s$:

$$p\left(\hat{C} = 1|s\right) = 0.5\lambda + (1 - \lambda)\left(\Phi(s; k, \sigma^2) - \Phi(s; -k, \sigma^2)\right) \tag{5}$$

Here, the additional parameter $\lambda$ reflected the probability of the observer lapsing, i.e., randomly guessing. This equation predicted the observer's response probabilities and could thus be fitted to a participant's behavioral responses.

Thus, for our first analysis regarding the relationship between the extracted IAF and the parameter values estimated for each participant, our BCI model had four free parameters: $p_{\text{same\_O}}$ and $p_{\text{same\_S}}$ were the prior probabilities of a common cause for vision and touch, independent of any sensory stimulation in the body ownership and simultaneity tasks, respectively; $\sigma$ was the noise impacting the measurement $x$; and $\lambda$ was a lapse rate to account for random guesses and unintended responses, which was expected to be very low and common to the two tasks (because both tasks were similar and tested in pseudorandom order). We ascertain the presence of a correlation between IAF and sensory uncertainty ($\sigma$), as well as between IAF and the prior probability of a common cause ($p_{\text{same\_O}}$ and $p_{\text{same\_S}}$), through Pearson and Spearman correlational analyses.

For our second analysis investigating which model parameters were impacted by tACS stimulation, we compared two versions of this model: in the first version, BCI_sigma, tACS stimulation was assumed to change the level of uncertainty in the measured asynchrony. This version of the model thus had six free parameters: $p_{\text{same\_O}}$ and $p_{\text{same\_S}}$ were the prior probabilities of a common cause for vision and touch, independent of any sensory stimulation in the body ownership and simultaneity tasks, respectively; $\sigma_{\text{low}}$, $\sigma_{\text{sham}}$, and $\sigma_{\text{high}}$ were the noises impacting the measurement $x$ under the different tACS conditions; and $\lambda$ was a lapse rate to account for random guesses and unintended responses. In the second version of BCI_psame, tACS stimulation was

assumed to change the a priori probability for the visual and tactile inputs originating from the same source. This version of the model thus had eight free parameters: $p_{same\_O\_low}$, $p_{same\_S\_low}$, $p_{same\_O\_sham}$, $p_{same\_S\_sham}$, $p_{same\_O\_high}$, and $p_{same\_S\_high}$ were the prior probabilities of a common cause for vision and touch, independent of any sensory stimulation in the body ownership and simultaneity tasks for each tACS condition, respectively; $\sigma$ was the noise impacting the measurement $x$; and $\lambda$ was a lapse rate to account for random guesses and unintended responses.

Model fitting was performed using maximum likelihood estimation implemented in MATLAB (MathWorks). We used the Bayesian adaptive direct search (BADS) algorithm[90], using 100 different initial parameter combinations per participant. The overall goodness of fit was assessed using the coefficient of determination, $R^2$, defined as

$$R^2 = 1 - \exp\left(-\frac{2}{n}\left(\log L(M) - \log L(M_0)\right)\right) \qquad (6)$$

where $\log L(M)$ and $\log L(M_0)$ denoted the log-likelihoods of the fitted and null models, respectively, and n was the number of data points[91]. For the null model, we assumed that an observer randomly chose one of the two response options, i.e., we assumed a discrete uniform distribution with a probability of 0.5. As the models' responses were discretized in our case to relate them to the two discrete response options, the coefficient of determination was divided by the maximum coefficient[91], defined as

$$\max\left(R^2\right) = 1 - \exp\left(\frac{2}{n}\log L(M_0)\right) \qquad (7)$$

The Akaike information criterion (AIC[64]) and Bayesian information criterion (BIC[65]) were used as measures of the goodness of fit of the model to compare the different versions of the model BCI_sigma and BCI_psame in the second analysis: the lower the AIC or BIC was, the better the fit. The BIC penalizes the number of free parameters more heavily than the AIC does. We calculated the AIC and BIC values for each model and participant according to the following equations:

$$AIC = 2n_{par} - 2\log L^* \qquad (8)$$

$$BIC = n_{trial}\log n_{par} - 2\log L^* \qquad (9)$$

where $L^*$ was the maximized value of the likelihood, $n_{par}$ was the number of free parameters, and $n_{trial}$ was the number of trials. We then calculated the differences in AIC and BIC between models and summed them across participants. We estimated a confidence interval by bootstrapping: 15 random AIC/BIC differences were drawn with replacement from the actual participants' AIC/BIC differences and summed; this procedure was repeated 10,000 times to compute the 95% CI. The model fit was performed on individual behavioral data, with parameter values estimated for each participant. The overall goodness of fit of the model, assessed using the coefficient of determination $R^2$, was very satisfactory[91] (Experiment 2: 98.8 +/- 1.6%, Fig. 8a, b; Experiment 3: BCI_sigma 97.1 +/- 2.6%, BCI_psame 97.2 +/- 2.5%), indicating that the model described the data well. We ascertained the presence of a correlation between IAF and sensory uncertainty, as well as IAF and prior probability of a common cause (Psame), through Pearson and Spearman correlational analyses.

## Reporting summary
Further information on research design is available in the Nature Portfolio Reporting Summary linked to this article.

## Data availability
The aggregated psychophysical and preprocessed EEG data generated in this study have been deposited in the Open Science Framework (OSF: https://osf.io/ytga5). Source data are provided in this paper Source data are provided with this paper.

## Code availability
The code for the EEG analysis has been deposited in the Open Science Framework (OSF: https://osf.io/ytga5/overview), as well as the code for the Bayesian computational modeling (OSF: https://osf.io/s5p4v/overview)

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

## Acknowledgements

We would like to thank Martti Mercurio for his help in building the robots and writing the program to control them and Mattias Karlén for creating the illustrations used in Figs. 1, 3, and 5. H.H.E. was supported by the European Research Council 2020 research and innovation program (SELF-UNITY #787386), Göran Gustafsson's foundation, and a Swedish Research Council Distinguished Professor grant (VR; #2017-03135). M.D. was supported by H2021 Marie Skłodowska-Curie Actions (Grant agreement no 101063812) and by Sweden's Innovation Agency (VINNOVA; #2022-01441). R.C.L. was supported by a Swedish Research Council (VR; #2024-00839) research grant and by a Strategic Research Area Neuroscience (StratNeuro) research fellowship. M.C. was supported by the French government under the France 2030 investment plan, as part of the Initiative d'Excellence d'Aix-Marseille Université – AMIDEX, AMX-23-CEI-030

## Author contributions

M.D. and H.H.E. conceived and conceptualized the study. M.D. collected the data, analyzed the data and created the figures. R.C.L. created the code for EEG data analysis. M.C. created the code for the computational modeling. R.C.L. and M.C. analyzed the data. M.D. and H.H.E. wrote the manuscript. All authors edited the manuscript.

## Funding

## Competing interests

The declare no competing interests.
