## [Transparent Peer Review file · Nature Communications]

Parietal alpha frequency shapes own-body perception by modulating the temporal integration of bodily signals

Corresponding Author: Dr Mariano D'Angelo

Version 0:

Reviewer comments:

Reviewer #1

(Remarks to the Author)

The article by D'Angelo et al. presents an innovative approach to understanding how individual alpha frequency (IAF) in the parietal cortex relates to both the temporal binding window (TBW) and perceptual sensitivity during body ownership and visuotactile simultaneity tasks. The findings indicate that higher IAF is associated with a narrower TBW and enhanced perceptual sensitivity, while lower frequencies produce the opposite effects. By employing a multi-method approach that combines rigorous experimental paradigms with computational modeling, the authors associate the observed effect with sensitivity while effectively discarding the alternative hypothesis of a bias effect, thereby avoiding potential spurious correlations between IAF and behavioral measures. This approach is particularly valuable given the methodological challenges raised in recent studies (Burgers and Noppeney, *Nature Human Behaviour*, 2022).

Furthermore, the work extends and is in line with influential research showing that IAF modulates decisional sensitivity (Di Gregorio et al., *Current Biology*, 2023; Samaha et al., *Current Biology*, 2015; Tarasi and Romei, *Journal of Cognitive Neuroscience*, 2023; Menetrey et al., *Imaging Neuroscience*, 2024; Nelli et al., *Nature Communications*, 2017) by demonstrating that a higher IAF also positively impacts body ownership perception. The authors further substantiate the causal link between IAF and temporal integration through brain stimulation experiments, which strengthen their claims. Additionally, by carefully controlling for the effects of alpha power, the authors align their findings with recent literature that emphasizes perceptual precision does not depend on alpha power (Tarasi et al., *Progress in Neurobiology*, 2022; Samaha et al., *Consciousness and Cognition*, 2022).

Overall, the study is well-designed and executed. Its clear theoretical grounding, robust experimental methodology, and comprehensive analytical approach contribute significantly to the field by reinforcing and expanding our understanding of the neural mechanisms underlying sensory integration and body perception. Based on its methodological rigor and impactful findings, I am fully in favor of its publication.

(Remarks on code availability)

Reviewer #2

(Remarks to the Author)

This paper by D'Angelo et al. shows that individual alpha frequency (IAF) from parietal cortex correlates with the variability of integration of visuotactile signal in the temporal dimension during a simultaneity judgment task and during the rubber hand illusion. This evidence is built via coherent psychophysics, electrophysiology, and computational results.

The study is well conceived, the paper well written, the results sometime spectacular. We have a few major points that deserve attention.

1. Source of the IAF modulation. The results from both EEH and the TACS experiments converge in pointing to the posterior parietal cortex. However, to conclude that the effect is specific for that area, the authors should investigate whether IAF from other regions correlate with the temporal effect. In particular, other regions are candidates for a potential role, e.g., the

ventral premotor cortex (VPMC), that is consistently activated during multisensory integrating and multisensory illusions. Actually, the same group has repetitively shown that VPMC activity correlates with ownership rating during the RHI and the body swap illusion (Ehrsson et al., *Science*, 2004; Brozzoli et al., *Journal of Neuroscience*, 2012; Petkova et al., *Current Biology* 2011). Another potential candidate is the TPJ, which is a multisensory hub, showing convergency of multisensory integration and modulation of bodily self-consciousness (Ionta et al., *Neuron*, 2011; Grivaz et al., *Neuroimage*, 2017). It would be important also to show that IAT from other areas do not correlate with the key elects. Similarly, while the use of parietal TACS is well justified, a control condition involving stimulation of another area would be necessary to conclusively demonstrate site-specific effects. We understand adding such controls may be practically unfeasible, and we do not think it is mandatory for publication. Still, the authors could moderate their claims regarding the specific role of the parietal cortex.

2. TACS Frequency Selection. We were somewhat surprised by the decision to stimulate at fixed frequencies of 8 Hz and 13 Hz, given that existing literature suggests that TACS (and other forms of rhythmic stimulation such as TMS) is most effective when applied at or near the individual alpha peak frequency (e.g., within ± 1 Hz). Although the results suggest that this approach was effective, additional methodological justification would be valuable. Specifically, the authors could elaborate on how stimulation frequencies were selected and discuss whether—and why—stimulation distant from the individual peak might still influence perception.

3. Is there any difference between the IAF from resting states and during the tasks?

I am a bit surprised to see no effect at all of alpha power. Several bodily self-consciousness effects have been linked to desynchronization of alpha / beta power. We understand this might not correlate with the TBW, but a rough difference between synchronous and largely asynchronous conditions during the task should be found.

4. Added Value of Computational Modelling. While the use of computational modelling to describe psychophysical data is valuable, we are not entirely convinced of its unique contribution in this specific context, especially given the emphasis placed on the model's ability to uncover underlying "mechanisms", for two main reasons. First, although the model successfully links temporal sensitivity and ownership modulations, these associations could plausibly have been established using purely behavioural or psychophysical methods. In terms of Marr's levels of analysis, the model contributes at the "algorithmic" level, offering a formal description of the integration process. However, it provides limited insight at the neural "implementation" level, which would be expected in a study focused on neural oscillations. Arguably, the main added value of the model in relation to alpha oscillations is that it shows IAF is linked to fitted temporal precision sigma. Isn't this not particularly surprising, considering that the model's structure imposes that TBWs and sigmas should correlate, and previous results already showed that IAF and TBW correlate? We suggest the authors reconsider the framing of the model's role, tempering claims about its mechanistic insight and better justifying its inclusion in light of the study's electrophysiological focus.

5. We have also a more theoretical, provocative point. If the simultaneity and the ownership judgment correlate with each other, and both correlate so well with the IAF, how are we really sure that we are assessing two different phenomena, and not the same simpler mechanisms? This question might have a trivial and more interesting perspective.

Trivial: are we sure that participants are answering to the two different questions, and are not just adopting the same response criterion – synchronous or not? More interesting: if participants are actually performing two different tasks, can we identify mechanisms underlying body ownership from basic multisensory integration? Can we establish mechanistic relationships between them?

Minor points:

m1. We understand the importance of controlling for bias in the TBW, by measuring sensitivity. We guess the two measures largely correlate. If so, we feel redundant always reporting the correlation between IAF and both measures. We would pick up one, and show once it works also for the other and put these results in the supplementary material.

m2. For the first finding of a correlation between TBW in the visuotactile task and during the RHI, the authors should cite the work of Francesca Ferri and Marcello Costantini, e.g., : Costantini M, Robinson J, Migliorati D, Donno B, Ferri F, Northol G. Temporal limits on rubber hand illusion reflect individuals' temporal resolution in multisensory perception. *Cognition*. 2016 Dec;157:39-48. doi: 10.1016/j.cognition.2016.08.010.

m3. Figure 4: b: the legend refers to the left parietal cortex. Is this the case? Not for the right? In the main text it seems the effect is bilateral.

m4. At times, the manuscript seems to treat phenomenal binding and multisensory integration (e.g., p.3, line 4; p.18, line 18) as synonyms. We think a clearer distinction between these concepts would strengthen the theoretical framing.

m5. We are not sure the analysis could be successful with the present data, but did the authors examine whether within participant fluctuations in IAF also predict TBWs or body ownership ratings?

m6. This is more of a suggestion as its applicability depends on the specific data. The use of a Gaussian fit to derive TBWs is standard, but may be suboptimal when participants show low temporal sensitivity. In such cases, the simultaneity perception curve could plateau broadly around zero as responses saturate. The authors might consider whether alternative fitting functions could yield better results.

m7. Adding power spectra/simultaneity judgements for individual participants in the SM may be useful.

(Remarks on code availability)

Version 1:

Reviewer comments:

Reviewer #1

(Remarks to the Author)

The authors have addressed my previous points. I do not have further comments and I congratulate for the excellent work.

(Remarks on code availability)

Reviewer #2

(Remarks to the Author)

The authors made an excellent job in revising the paper.

I appreciate the open scientific exchanges on the different points.

I believe adding the results from the premotor cortex adds interesting information. The fact that IAF from the premotor cortex correlated with body ownership / TBW for body-related stimuli but not with temporal binding windows for simultaneity judgements is very interesting. In the discussion, page 21-22, the quickly discuss this point:

"The apparent selectivity of premotor IAF for body ownership aligns well with previous literature showing that activity in this area is related to body ownership beyond visuotactile synchrony/asynchrony effects^{34,37} ; RHI-related PPC activity is closely linked to the individual visuotactile stimulations delivered to participants' real and rubber hands, whereas premotor activity is more sustained between individual touches and reflects the between IAF and temporal integration continuous feeling of hand ownership"

The final part of the sentence is not 100% clear to me - in which sense premotor activity is more sustained, and what is the result supporting this claim? To me the dissociation between premotor and parietal results are interesting, and can be framed in a more general way, i.e., the dissociations between "higher-level" experience-related encoding (body ownership) and and lower-level sensory integration tasks. By the way this rules out also our previous concern about the specificity of the tasks vs. the objection of th participants responding in the same way to any task (our previous point 5)

Said that, it would be fair to report also an analyses from a control area showing no correlation with the tasks at all to demonstrate specificity of the effects. At least in the supplementary materials.

Besides these minor points, the paper is suitable for publications for me.

Once again congratulations to the authors for this excellent study, and thanks for taking into account properly all our comments.

Andrea Serino

(Remarks on code availability)

Reviewer #1 (Remarks to the Author):

The article by D'Angelo et al. presents an innovative approach to understanding how individual alpha frequency (IAF) in the parietal cortex relates to both the temporal binding window (TBW) and perceptual sensitivity during body ownership and visuotactile simultaneity tasks. The findings indicate that higher IAF is associated with a narrower TBW and enhanced perceptual sensitivity, while lower frequencies produce the opposite effects. By employing a multi-method approach that combines rigorous experimental paradigms with computational modeling, the authors associate the observed effect with sensitivity while effectively discarding the alternative hypothesis of a bias effect, thereby avoiding potential spurious correlations between IAF and behavioral measures. This approach is particularly valuable given the methodological challenges raised in recent studies (Buergers and Noppeney, Nature Human Behaviour, 2022).

Furthermore, the work extends and is in line with influential research showing that IAF modulates decisional sensitivity (Di Gregorio et al., Current Biology, 2023; Samaha et al., Current Biology, 2015; Tarasi and Romei, Journal of Cognitive Neuroscience, 2023; Menetrey et al., Imaging Neuroscience, 2024; Nelli et al., Nature Communications, 2017) by demonstrating that a higher IAF also positively impacts body ownership perception. The authors further substantiate the causal link between IAF and temporal integration through brain stimulation experiments, which strengthen their claims. Additionally, by carefully controlling for the effects of alpha power, the authors align their findings with recent literature that emphasizes perceptual precision does not depend on alpha power (Tarasi et al., Progress in Neurobiology, 2022; Samaha et al., Consciousness and Cognition, 2022).

Overall, the study is well-designed and executed. Its clear theoretical grounding, robust experimental methodology, and comprehensive analytical approach contribute significantly to the field by reinforcing and expanding our understanding of the neural mechanisms underlying sensory integration and body perception. Based on its methodological rigor and impactful findings, I am fully in favor of its publication.

Reply: We thank the reviewer for the thoughtful and encouraging comments. We are pleased that they appreciated our multi-method approach and the theoretical grounding of the study. We also thank the reviewer for acknowledging the relevance of our findings within the current literature. These remarks are very motivating, and we appreciate the reviewer for his support towards publication.

Reviewer #2 (Remarks to the Author):

This paper by D'Angelo et al. shows that individual alpha frequency (IAF) from parietal cortex correlates with the variability of integration of visuotactile signal in the temporal dimension during a simultaneity judgment task and during the rubber hand illusion. This evidence is built via coherent psychophysics, electrophysiology, and computational results.

The study is well conceived, the paper well written, the results sometime spectacular. We have a few major points that deserve attention.

Reply: We thank the reviewers for their appreciation of the study design, clarity of writing, and the combination of psychophysics, EEG, and computational modelling. We welcome the opportunity to address the major points raised below.

1. Source of the IAF modulation. The results from both EEH and the TACS experiments converge in pointing to the posterior parietal cortex. However, to conclude that the effect is specific for that area, the authors should investigate whether IAF from other regions correlate with the temporal effect. In particular, other regions are candidates for a potential role, e.g., the ventral premotor cortex (VPMC), that is consistently activated during multisensory integrating and multisensory illusions. Actually, the same group has repetitively shown that VPMC activity correlates with ownership rating during the RHI and the body swap illusion (Ehrsson et al., Science, 2004; Brozzoli et al., Journal of Neuroscience, 2012; Petkova et al., Current Biology 2011). Another potential candidate is the TPJ, which is a multisensory hub, showing convergency of multisensory integration and modulation of bodily self-consciousness (Ionta et al., Neuron, 2011; Grivaz et al., Neuroimage, 2017). It would be important also to show that IAF from other areas do not correlate with the key effects. Similarly, while the use of parietal TACS is well justified, a control condition involving stimulation of another area would be necessary to conclusively demonstrate site-specific effects. We understand adding such controls may be practically unfeasible, and we do not think it is mandatory for publication. Still, the authors could moderate their claims regarding the specific role of the parietal cortex.

Reply: We thank the reviewers for giving us the opportunity to clarify our ROI selection. Our study was designed to test the involvement of the posterior parietal cortex (PPC) in a hypothesis-driven manner, based on a well-established body of literature linking PPC activity to visuotactile integration and body ownership. We clarified this in our Methods section (page 9, lines 10-12):

“We focused on IAF in the posterior parietal cortex because previous fMRI studies have implicated this region in multisensory integration underlying body ownership³³⁻⁴¹, and parietal IAF is readily recorded during visuotactile paradigms. Thus, we adopted a hypothesis-driven approach focusing on the PPC, where we hypothesized IAF to play a central role in visuotactile integration.”

Our aim was not to claim that the observed IAF effects are unique to PPC, but rather to test whether alpha oscillations in the PPC are functionally related to individual differences in visuotactile temporal integration that support the sense of body ownership. In the discussion section, we clarified that we do not claim that the observed IAF are necessarily unique to PPC, and that we might observe similar IAF effects in other multisensory areas involved in body ownership, such as the ventral premotor cortex (page 21, lines 24-34, page 22, lines 1-3). Moreover, we have now conducted an exploratory post-hoc analysis considering the IAF measured over the premotor cortex, as reviewers suggested.

“However, we do not claim that the PPC is the only cortical region where IAF may influence visuotactile integration. Similar mechanisms may operate in other multisensory areas, such as the ventral premotor cortex, which has been particularly implicated in the feeling of body ownership.^{34,40} An exploratory post-hoc analysis revealed that IAF over the premotor cortex showed significant correlations with body ownership TBWs and sensitivities (Supplementary Results for Experiment 2; Fig. S6), but not with the corresponding visuotactile simultaneity measures (Supplementary Results for Experiment 2). This finding suggests that the role of IAF in shaping temporal perception extends beyond the PPC, at least for body ownership. The apparent selectivity of premotor IAF for body

ownership aligns well with previous literature showing that activity in this area is related to body ownership beyond visuotactile synchrony/asynchrony effects^{34,37}; RHI-related PPC activity is closely linked to the individual visuotactile stimulations delivered to participants' real and rubber hands, whereas premotor activity is more sustained between individual touches and reflects the continuous feeling of hand ownership⁴²”

We have mentioned the post-hoc analysis considering the IAF measured over the premotor corte in the main manuscript (page 12, lines 13-16, page 13, lines 1-6):

“Experiment 2: post-hoc premotor IAF analyses

*In a separate exploratory post-hoc analysis, we also analyzed frontal electrodes corresponding to the premotor cortex (see Supplementary Results for Experiment 2), another multisensory area strongly implicated in body ownership experience^{34,40}, to investigate whether the IAF PPC findings are unique to this area or generalize to other regions involved in body ownership. We found that both task-related and resting state premotor IAF from either hemisphere correlated significantly with body ownership sensitivity and body ownership TBW (all $p < 0.05$; Supplementary Results for Experiment 2; **Fig. S6**). The IAF correlations with the corresponding visuotactile simultaneity measures were mostly nonsignificant, with the exception of a significant correlation between resting state IAF in the right hemisphere and simultaneity sensitivity.”*

We have added the results of this new analysis in the Supplementary Results for Experiment 2 (page 13, lines 19-23; page 14 lines 1-22; page 15, lines 1-15, page 16, lines 1-9).

“We conducted an exploratory post-hoc analysis to examine whether FOOOF derived IAF extracted from electrodes corresponding to the premotor cortex would show a similar pattern of correlations as observed for parietal IAF. This analysis was motivated by evidence implicating the premotor cortex, particularly its ventral portion, in the experience of body ownership^{9,10}. To this end, we selected two frontal ROIs corresponding to electrodes over the premotor cortex, including its ventral regions: D7, D6, D5, D4, D3, C24, C32, C26, C25 (left hemisphere) and C11, C3, C4, C5, C6, C7, C12, C13, C10 (right hemisphere). FOOOF successfully identified IAF for the majority of the 18 selected electrodes during the resting state (mean = 17.087; range = 10–18), as well as during the body ownership task (mean = 15.391; range = 2–18) and the visuotactile simultaneity judgments (mean = 15.652; range = 4–18). However, the algorithm failed to detect a reliable alpha peak in the left hemisphere of one participant during both task conditions. Consequently, this participant's data were excluded from the task-related IAF analyses.

We found that task-related IAF over the premotor cortex was significantly correlated with body ownership TBW, both in the left hemisphere (Pearson's $r = -.475$, $p = .001$; Spearman's $\rho = -.468$, $p = .001$; $N = 45$) and the right hemisphere (Pearson's $r = -.469$, $p = .001$; Spearman's $\rho = -.421$, $p = .004$; $N = 45$). A similar pattern was observed for body ownership sensitivity, with significant correlations in both the left (Pearson's $r = .606$, $p < .001$; Spearman's $\rho = .624$, $p < .001$; $N = 45$) and right hemisphere (Pearson's $r = .652$, $p < .001$; Spearman's $\rho = .648$, $p < .001$; $N = 45$). In contrast, no significant correlations were found between task-related IAF and visuotactile simultaneity TBW, either in the left hemisphere (Pearson's $r = -.182$, $p = .233$; Spearman's $\rho = -.106$, $p = .487$; $N = 45$) or the right hemisphere (Pearson's $r = -.269$, $p = .074$; Spearman's $\rho = -.203$, $p = .181$; $N = 45$). This absence of correlation was further confirmed when considering simultaneity sensitivity, which showed no significant association with IAF in either the left (Pearson's $r = .121$, $p = .427$; Spearman's $\rho = .090$, $p = .555$; $N = 45$) or right hemisphere (Pearson's $r = .237$, $p = .116$; Spearman's $\rho = .210$, $p = .166$; $N = 45$).

Correlation between body ownership and task-related PMC IAF

Correlation between body ownership and resting state PCM IAF

Fig. S6 | Correlations between task-related and resting state FOOOF-Individual Alpha Frequency (IAF) in the left premotor cortex (PMC) and body ownership. **a**, Correlation between task-related IAF over the left PMC (i.e., the hemisphere contralateral to the visuotactile stimulation) and body ownership TBW. **b**, Correlation between task-related IAF over the left PMC and body ownership sensitivity. **c**, Correlation between resting state IAF over the left PMC and body ownership TBW **d**, Correlation between resting state IAF over the left PMC and body ownership sensitivity. The shaded region reflects the 95% confidence interval

When examining resting-state IAF over the premotor cortex, we again found significant correlations with body ownership TBW, in both the left (Pearson's $r = -.408$, $p = .005$; Spearman's $\rho = -.431$, $p = .003$; $N = 46$) and right hemispheres (Pearson's $r = -.487$, $p = .001$; Spearman's $\rho = -.521$, $p < .001$; $N = 46$). The same pattern held for body ownership sensitivity, with significant correlations in the left (Pearson's $r = .434$, $p = .003$; Spearman's $\rho = .429$, $p = .003$; $N = 46$) and right hemispheres (Pearson's $r = .462$, $p = .001$; Spearman's $\rho = .516$, $p < .001$; $N = 46$). As for visuotactile simultaneity, results were mixed. We observed a trend toward significance in the left premotor cortex (Pearson's $r = -.290$, $p = .050$; Spearman's $\rho = -.237$, $p = .112$; $N = 46$), and a significant correlation in the right hemisphere (Pearson's $r = -.303$, $p = .041$; Spearman's $\rho = -.297$, $p = .049$; $N = 46$).

Taken together, these findings suggest that premotor IAF is involved in the temporal integration of visuotactile signals related to the sense of body ownership. In contrast, no such relationship was observed for visuotactile simultaneity and task-related IAF, and the relationship between resting-state IAF and simultaneity was inconsistent. Thus, PPC IAF does not have a unique role in temporal multisensory perception, as IAF can modulate temporal multisensory perception in different brain regions in a task-relevant manner. Specifically, premotor IAF appears to have a more selective role in body ownership temporal perception than PPC IAF."

2. *TACS Frequency Selection. We were somewhat surprised by the decision to stimulate at fixed frequencies of 8 Hz and 13 Hz, given that existing literature suggests that TACS (and other forms of rhythmic stimulation such as TMS) is most effective when applied at or near the individual alpha peak frequency (e.g., within ± 1 Hz). Although the results suggest that this approach was effective, additional methodological justification would be valuable. Specifically, the authors could elaborate on how stimulation frequencies were selected and discuss whether—and why—stimulation distant from the individual peak might still influence perception.*

Reply: We thank the reviewers for the opportunity to clarify the rationale behind our tACS experiment. We acknowledge that 8 Hz and 13 Hz lie at the lower and upper boundaries of the canonical alpha band and may differ from participants' IAF. However, our aim was not to increase oscillatory power or alpha amplitude through phase pulling—effects that typically require stimulation at the individual alpha peak. Instead, we sought to test whether stimulation at faster or slower frequencies, distinctly below or above the IAF, could causally influence visuotactile temporal integration by modulating the frequency of endogenous oscillatory activity toward faster or slower regimes. We have now added a small paragraph in the Methods section to motivate our frequency choice (page 13, lines 19-25):

“EEG Experiment 2 had revealed that the IAF ranged between 9.17 and 12.33 Hz during the perceptual judgment tasks after controlling for the aperiodic component. We therefore selected 8 Hz and 13 Hz frequencies to ensure that, across participants, stimulation frequencies would consistently fall outside their endogenous alpha range. This allowed us to exogenously shift the frequency of ongoing alpha oscillations toward slower or faster rhythms via entrainment, thereby modulating the temporal resolution of perceptual processing during the tasks.”

We agree with the reviewers that external rhythmic input may be more effective at shifting intrinsic frequencies and eliciting strong behavioral effects when applied in close proximity to the endogenous rhythm. Accordingly, individualized stimulation at IAF ± 1 Hz (di Gregorio et al., 2022) or ± 2 Hz (Cecere et al., 2015) would likely have enhanced the behavioral effect. We now acknowledge this in the discussion section (page 22, lines 23-34):

“Regarding tACS experiment, it is important to highlight the substantial inter-individual variability in stimulation effects, ranging from no observable effect to marked behavioral changes. This variability may reflect differences in how stimulation at the edges of the alpha band (8 Hz and 13 Hz) interacted with each participant's intrinsic alpha peak. It is therefore possible that targeting frequencies closer to the IAF (e.g., $\pm 1-2$ Hz) would have yielded stronger behavioral effects^{11,14,53}. Nevertheless, stimulation at 8 Hz and 13 Hz was still effective in producing significant group-level modulation, consistent with previous tACS studies that applied stimulation near the boundaries of the target frequency band^{54,88}. For example, Venskus et al. (2022) applied tACS at low (8 Hz) and high (14 Hz) alpha frequencies and observed significant modulation of the temporal binding window associated with the sense of agency. Similarly, Wolinski et al. (2018) applied theta-band stimulation (4 Hz and 7 Hz) and reported corresponding changes in visuospatial working memory capacity”

3. *Is there any difference between the IAF from resting states and during the tasks? I am a bit surprised to see no effect at all of alpha power. Several bodily self-consciousness effects have been linked to desynchronization of alpha / beta power. We understand this might not correlate*

with the TBW, but a rough difference between synchronous and largely asynchronous conditions during the task should be found.

Reply: To address the reviewers' question regarding potential differences in IAF between the perceptual tasks and the resting state, we conducted a 3×2 ANOVA on IAF, with Condition (Ownership judgments, Simultaneity judgments, Resting state) and Hemisphere (Left, Right) as factors. This analysis revealed a significant main effect of Condition ($F_{2,90} = 3.125$; $p = 0.049$; $\eta_p^2 = 0.062$). Importantly, we observed the same main effect when using FOOOF-derived IAF values ($N = 35$; $F_{2,68} = 3.757$; $p = 0.028$; $\eta_p^2 = 0.099$). However, in both cases, none of the pairwise comparisons survived Holm-Bonferroni correction for multiple comparisons. This suggests that the overall effect of condition on IAF is present but not driven by strong or consistent differences between specific pairs of conditions.

Regarding the reviewers' interesting comment on alpha power, we believe this is a particularly relevant point in light of the ongoing debate about the respective roles of alpha frequency and power in perception. Recent literature suggests that perceptual precision and accuracy are more strongly associated with alpha frequency than with alpha power. Our findings align well with this perspective, indicating that IAF, but not alpha power, predicts the temporal resolution of multisensory integration underlying body ownership. That said, we fully agree with the reviewers that previous studies have consistently reported reductions in alpha power during bodily illusions, particularly in multisensory regions such as the posterior parietal cortex. However, alpha desynchronization in this context reflects the activation of multisensory brain areas, rather than being a direct marker of perceptual temporal precision.

To address the reviewers' concern on alpha power, we compared it during body ownership judgments, simultaneity judgments, and resting state (page 11, lines 3-6):

“In addition, given that bodily illusions are typically associated with a reduction in alpha power^{52,53}, we conducted a sanity check to confirm that alpha power differed between the resting state and the perceptual tasks (Supplementary Results for Experiment 2).”

The results can be found in the Supplementary Results of Experiment 2 (page 5 lines 29-34, page 6 lines 1-5):

“Bodily illusions are typically associated with a reduction in alpha power. To examine whether this was also the case in our data, we conducted a 3×2 ANOVA on alpha power, with Condition (Ownership judgments, simultaneity judgments, resting state) and Hemisphere (Left, Right) as factors. This analysis aimed to test whether alpha power was reduced during the perceptual tasks compared to the resting state. We indeed found a main effect of condition ($F_{2,90} = 13.879$; $p < 0.001$; $\eta_p^2 = 0.236$) such that the alpha power was lower during body ownership ($t = -3.792$; $p < .001$) and simultaneity judgments ($t = -3.753$; $p < .001$) as compared to the resting state. This result supports previous findings showing that bodily illusions and active multisensory processing are associated with alpha desynchronization, likely reflecting increased cortical excitability and engagement of multisensory areas.”

As suggested by the reviewers, we also examined whether alpha power differed between the synchronous condition (0 ms) and the large asynchronous conditions (−400 ms and +400 ms). To this end, we ran two ANOVAs on alpha power, with Asynchrony (0 ms, −400 ms, +400 ms) and Hemisphere as factors, separately for the body ownership and simultaneity judgment tasks. However, we found no main effect of the asynchronies in both body ownership ($F_{2,90} = 2.113$; $p = 0.127$; $\eta_p^2 =$

0.045) and simultaneity judgments ($F_{2,90} = 1.578$; $p = 0.212$; $\eta_p^2 = 0.034$), neither a significant interaction between asynchronies and hemispheres. However, since our paradigm was not specifically designed for this comparison, these null results should be interpreted with caution. Dividing the trials into three asynchrony conditions reduced the number of observations contributing to each condition, thereby limiting statistical power to detect possible effects.

4. Added Value of Computational Modelling. While the use of computational modelling to describe psychophysical data is valuable, We are not entirely convinced of its unique contribution in this specific context, especially given the emphasis placed on the model's ability to uncover underlying "mechanisms", for two main reasons. First, although the model successfully links temporal sensitivity and ownership modulations, these associations could plausibly have been established using purely behavioural or psychophysical methods. In terms of Marr's levels of analysis, the model contributes at the "algorithmic" level, offering a formal description of the integration process. However, it provides limited insight at the neural "implementation" level, which would be expected in a study focused on neural oscillations. Arguably, the main added value of the model in relation to alpha oscillations is that it shows IAF is linked to fitted temporal precisions sigma. Isn't this not particularly surprising, considering that the model's structure imposes that TBWs and sigmas should correlate, and previous results already showed that IAF and TBW correlate? We suggest the authors reconsider the framing of the model's role, tempering claims about its mechanistic insight and better justifying its inclusion in light of the study's electrophysiological focus.

Reply: We thank the reviewers for this thoughtful comment, which prompted us to clarify the scope and purpose of the computational modelling approach used in our study. We believe the model provides added value that goes beyond the descriptive use of the TBW. While the TBW—defined in our study as the standard deviation of the Gaussian fit—indicates the temporal interval within which two stimuli are likely to be integrated, it does not provide insights into the underlying processes that drive this integration. By contrast, our Bayesian model allows us to formally disambiguate between two distinct computational components: (1) the prior probability that two sensory signals originate from a common cause, and (2) the sensory uncertainty (σ) associated with the encoding of those signals. This decomposition is critical, as it enables us to infer whether a broader TBW reflects an increase in sensory uncertainty, a shift in prior expectations, or a combination of both. Such a distinction cannot be made from behavioral data alone without a model-based approach.

While prior research has shown correlations between IAF and TBW, our findings specify that IAF is associated with sensory uncertainty (e.g., the inverse of sensory precision). We believe that this relationship is not trivial, as it better explains the role of IAF in the probabilistic perceptual inference about the causes of the sensory signals, suggesting that alpha oscillations may dictate the intrinsic reliability of the perceptual system, rather than being a byproduct of multisensory integration or attentional engagement. While we agree that our Bayesian model operates at the algorithmic level in Marr's hierarchy, and that EEG measures such as IAF pertain to the level of large-scale neural implementation, we believe that linking these two levels is precisely the goal of computational neuroscience. In fact, establishing such bridges between formal models and neural signals is a widely accepted and productive strategy in cognitive neuroscience. A well-established example is model-based fMRI, where internal variables from a computational model are correlated with BOLD responses in specific brain regions (e.g., Gläscher and O'Doherty 2010). Another example is Rohe et al 2019 who combined EEG with Bayesian causal inference modelling to find that prestimulus oscillatory alpha power and phase correlate with observers' prior beliefs (p_{same}). Thus, this

approach of linking neural measures to ‘abstract’ computations in models is a well-established approach, even if the precise microcircuit-level implementation remains unknown in most cases. In the same spirit, our study links a computational parameter, sensory uncertainty (σ), to a neural marker, IAF. Although alpha frequency is a macroscopic measure, it reflects constraints on sensory precision, making it a meaningful implementation-level correlate of the algorithmic computation modeled in our framework. Thus, we believe our approach fits well within the logic of computational neuroscience: bridging behavioral performance, formal models, and neural dynamics.

That said, we fully agree with the reviewers on the importance of tempering any claims that might be interpreted as suggesting that the model reveals mechanistic neural implementation in a strict sense. Accordingly, we have carefully revised the manuscript to reflect this. Specifically, we have removed references to “mechanisms” in the abstract and modified several key statements throughout the text. For example, in the Results section, the model is introduced as follows (page 16, lines 15-18):

“Building on Experiments 2 and 3, we applied a probabilistic computational model based on Bayesian causal inference to quantify how individual differences in IAF influence perceptual judgments of ownership and simultaneity, offering a principled computational framework to explain the observed behavioral patterns.”

We also revised a related sentence to clarify the interpretative rather than mechanistic scope of the results (page 18, lines 17-21):

“This result aligns well with the findings on how IAF is related to TBWs and sensitivities, while offering a deeper computational understanding of how alpha oscillations may constrain the temporal resolution of sensory integration via their relationship with internal sensory uncertainty.”

Similarly, in the Discussion section, we have eliminated all references to mechanistic interpretations of the model. These revisions were made to ensure conceptual consistency across the manuscript, and to avoid overinterpreting the model's mechanistic implications (page 20, lines 5-7; page 22, lines 16-18).

“Our computational modeling results provide insight into how IAF may shape the temporal integration of multisensory signals by modulating sensory uncertainty involved in perceptual inference”

“(…) Second, computational modeling supported our conclusions and provided a formal account of how IAF modulates perception through its relationship with sensory uncertainty.”

5. We have also a more theoretical, provocative point. If the simultaneity and the ownership judgment correlate with each other, and both correlate so well with the IAF, how are we really sure that we are assessing two different phenomena, and not the same simpler mechanisms? This question might have a trivial and more interesting perspective.

Trivial: are we sure that participants are answering to the two different questions, and are not just adopting the same response criterion – synchronous or not? More interesting: if participants are actually performing two different tasks, can we identify mechanisms underlying

body ownership from basic multisensory integration? Can we establishing mechanistic relationships between them?

Reply: We thank the reviewer for raising this important theoretical point. Regarding the more “trivial” concern, we observed consistent and significant differences between body ownership and simultaneity judgments across all experiments, both in terms of the width of the TBW and in terms of their perceptual sensitivity; and these differences are clearly seen at the individual subject level. Given that the two tasks differed at the behavioural level and were presented in separate blocks with distinct instructions, we consider it virtually impossible that participants were treating them as equivalent. In the previous version of the manuscript, we reported the comparison between body ownership and simultaneity TBWs in Experiment 1. We have now added the comparison between body ownership and simultaneity sensitivity as well (page 8, lines 16-18).

“Moreover, consistently with TBW results, sensitivity to body ownership was lower as compared to sensitivity to visuotactile simultaneity ($t = -12.44$; $p < .001$; $d = -2.272$)”

We have also included all comparisons between TBWs and sensitivities across both tasks for Experiment 2 and 3 in the Supplementary Results with a short reference to these results in the main manuscript (page 11, lines 7-9; page 11, lines 31-32; page 16, lines 2-3).

As for the more theoretical and thought-provoking point raised by the reviewers, we agree that visuotactile simultaneity detection is an important component of body ownership. In fact, this was the starting premise of our study: that body ownership relies, at least in part, on the detection of temporal congruence across sensory modalities. The observed correlations with IAF capture what these two phenomena—simultaneity perception and body ownership—have in common: the temporal resolution of the perceptual system. However, the fact that both processes are modulated by a shared electrophysiological marker, namely, IAF, does not imply that they are functionally equivalent. Rather, it suggests that IAF constrains a common temporal resolution mechanism that supports both simultaneity perception and the temporal aspects of multisensory integration involved in body ownership. Importantly, we believe that the body ownership task is not reducible to the simultaneity task, as it involves a more complex form of multisensory integration. This includes not only visuotactile but also visuoproprioceptive signals. Crucially, in body ownership judgments multisensory integration operates not only in the temporal domain but also in the spatial domain (Samad et al 2015; Chancel et al 2023; Lanfranco et al 2023; Fang et al 2019), requiring the integration of signals that are spatially aligned with the body and coherent with internal body representations. This adds a layer of complexity to the perceptual inference processes underlying the experience of ownership.

Identifying the specific neural mechanisms that distinguish body ownership from basic visuotactile integration is indeed an important and challenging question. However, this goes beyond the scope of the present study, which aimed instead to highlight what these two processes have in common—namely, their dependence on the temporal resolution afforded by the speed of alpha oscillations. Of course, many previous fMRI and ECoG studies have distinguished body ownership effects from basic visuotactile synchrony effects by including control conditions with synchronous versus asynchronous visuotactile stimulation in factorial designs (e.g., Ehrsson et al., 2004; Gentile et al., 2013; Guterstam et al 2019), demonstrating that neural substrates of body ownership are distinct from basic visuotactile integration.

Minor points:

m1. We understand the important of controlling for bias in the TBW, by measuring sensitivity. We guess the two measures largely correlate. If so, we feel redundant always reporting the correlation between IAF and both measures. we would pick up one, and show once it works also for the other and put these results in the supplementary material.

Reply: We understand the reviewers' concern that including both measures might appear redundant. However, we chose to report both the correlations between IAF and TBW and between IAF and sensitivity in the main manuscript for theoretical and methodological reasons. Previous studies that reported associations between IAF and TBW have been criticized—particularly by Buergers and Noppeney (2022)—for relying on heterogeneous behavioral definitions of the TBW and for not controlling for potential response bias. These critiques have called for more rigorous psychophysical approaches, including the use of signal detection theory. By including both measures in the manuscript, we aim to strengthen the robustness of our findings and to align with current methodological recommendations in the field.

m2. For the first finding of a correlation between TBW in the visuotactile task and during the RHI, the authors should cite the work of Francesca Ferri and Marcello Costantini, e.g., : Costantini M, Robinson J, Migliorati D, Donno B, Ferri F, Northo I G. Temporal limits on rubber hand illusion reflect individuals' temporal resolution in multisensory perception. Cognition. 2016 Dec;157:39-48. doi: 10.1016/j.cognition.2016.08.010.

Reply: We thank the reviewers for the suggestion. We have now cited the Costantini et al. (2016) paper in the introduction (page 4, lines 8-9).

“Body ownership depends on the multisensory integration of self-related signals from different sensory modalities, including vision, touch, and proprioception. Temporal discrepancies among these signals are critical in determining whether they should be integrated and whether body ownership is perceived²⁹⁻³².

m3. Figure 4: b: the legend refers to the left parietal cortex. Is this the case? Not for the right? In the main text it seems the elect is bilateral.

Reply: We thank the reviewers for pointing this out. As noted in the main text, the effect was indeed observed in both hemispheres. However, in the figure, we chose to display the left parietal cortex only, as it is contralateral to the site of visuotactile stimulation. We have now clarified this point in the figure legend to avoid confusion (page 12, line 5).

“Correlation between task-related IAF measured in the left posterior parietal cortex (i.e., the hemisphere contralateral to the stimulation) during body ownership judgment tasks and body ownership TBW”

m4. At times, the manuscript seems to treat phenomenal binding and multisensory integration (e.g., p.3, line 4; p.18, line 18) as synonyms. We think a clearer distinction between these concepts would strengthen the theoretical framing.

Reply: We thank the reviewer for this comment. While the terms "multisensory integration" and "multisensory binding" are often used interchangeably in the literature (e.g., in studies of the rubber hand illusion and visuotactile synchrony), we agree that "binding" may sometimes be

interpreted more narrowly, referring to the subjective unification of sensory features into a single percept. In contrast, "multisensory integration" is a broader term commonly used in neuroscience and psychophysics to describe how the brain combines information from different sensory modalities. We have reviewed all instances where we use the terms "binding" and "integration" to ensure that the most appropriate term is used in each context. Given the substantial overlap between these concepts in the field, however, we do not believe a strict theoretical distinction is necessary for the purposes of this manuscript.

m5. We are not sure the analysis could be successful with the present data, but did the authors examine whether within participant fluctuations in IAF also predict TBWs or body ownership ratings?

Reply: We thank the reviewer for raising this important question. We did not perform a trial-by-trial analysis to test whether IAF fluctuations predict participants' responses, as we believe our current experimental design is not optimal and sensitive enough for such an analysis. However, we agree that this is a relevant and challenging question that future studies could address more directly. To properly test this hypothesis, an experimental design with only two delay conditions—one at 0 ms and another at an individually tailored threshold—would be more appropriate (e.g., Di Gregorio et al., 2022). Such a design would allow for sufficient statistical power to evaluate trial-by-trial fluctuations in perceptual reports as a function of momentary IAF dynamics.

m6. This is more of a suggestion as its applicability depends on the specific data. The use of a Gaussian fit to derive TBWs is standard, but may be suboptimal when participants show low temporal sensitivity. In such cases, the simultaneity perception curve could plateau broadly around zero as responses saturate. The authors might consider whether alternative fitting functions could yield better results.

Reply: We thank the reviewers for raising this point. We agree that the optimal choice of fitting function may depend on the characteristics of the data, particularly when participants show low temporal sensitivity. However, in our dataset, the Gaussian fit provided a good approximation of participants' simultaneity perception curves. To support this, we have now included the corresponding R^2 values in the Methods section (page 25, lines 22-24; page 28, lines 29-31).

m7. Adding power spectra/simultaneity judgements for individual participants in the SM may be useful.

Reply: We thank the reviewers for this suggestion. We have made available, via OSF, the individual simultaneity and body ownership responses, as well as all individual values of IAF and alpha power (<https://osf.io/ytga5/>). This should allow interested readers to explore the relationship between spectral measures and behavioral responses at the individual participant level.

References

Buergers, S., & Noppeney, U. The role of alpha oscillations in temporal binding within and across the senses. *Nat. Hum Behav.* **6**, 732-742 (2022).

Chancel M, Ehrsson HH. Proprioceptive uncertainty promotes the rubber hand illusion. *Cortex*. 2023 Aug;165:70-85. doi: 10.1016/j.cortex.2023.04.005.

Di Gregorio, F., Trajkovic, J., Roperti, C., Marcantoni, E., Di Luzio, P., Avenanti, A., & Romei, V. Tuning alpha rhythms to shape conscious visual perception. *Curr. Biol.* **32**, 988-998 (2022).

Ehrsson, H. H., Spence, C., & Passingham, R. E. That's my hand! Activity in premotor cortex reflects feeling of ownership of a limb. *Science*. **305**, 875-877 (2004).

Fang W, Li J, Qi G, Li S, Sigman M, Wang L. Statistical inference of body representation in the macaque brain. *Proc Natl Acad Sci.* **116**, 20151-20157. (2019)

Gentile, G., Guterstam, A., Brozzoli, C., & Ehrsson, H. H. Disintegration of multisensory signals from the real hand reduces default limb self-attribution: an fMRI study. *J. Neurosci.* **33**, 13350-13366 (2013).

Gläscher, J. P., & O'Doherty, J. P. Model-based approaches to neuroimaging: combining reinforcement learning theory with fMRI data. *Wiley Interdiscip. Rev. Cogn. Sci.* **1**, 501-510. (2010).

Guterstam, A., Collins, K. L., Cronin, J. A., Zeberg, H., Darvas, F., Weaver, K. E., ... & Ehrsson, H. H. Direct electrophysiological correlates of body ownership in human cerebral cortex. *Cereb. Cortex.* **29**, 1328-1341 (2019).

Lanfranco RC, Chancel M, Ehrsson HH. Quantifying body ownership information processing and perceptual bias in the rubber hand illusion. *Cognition.* **238**, 105491 (2023)

Rohe, T., Ehlis, A. C., & Noppeney, U. The neural dynamics of hierarchical Bayesian causal inference in multisensory perception. *Nat Comm.* **10**, 1907, (2019)

Samad M, Chung AJ, Shams L. Perception of body ownership is driven by Bayesian sensory inference. *PLoS One.* **10**, e0117178. (2015)

Venskus, A., Ferri, F., Migliorati, D., Spadone, S., Costantini, M., & Hughes, G. Temporal binding window and sense of agency are related processes modifiable via occipital tACS. *PLoS One.* **16**, e0256987 (2021).

Wolinski, N., Cooper, N. R., Sauseng, P., & Romei, V. The speed of parietal theta frequency drives visuospatial working memory capacity. *PLoS Biol.* **16**, e2005348. (2018)

*The authors made an excellent job in revising the paper.
I appreciate the open scientific exchanges on the different points.*

Reply: We are glad that the reviewer appreciated our responses and the scientific exchange. We welcome the opportunity to answer these last minor points

I believe adding the results from the premotor cortex adds interesting information. The fact that IAF from the premotor cortex correlated with body ownership / TBW for body-related stimuli but not with temporal binding windows for simultaneity judgements is very interesting. In the discussion, page 21-22, the quickly discuss this point:

"The apparent selectivity of premotor IAF for body ownership aligns well with previous literature showing that activity in this area is related to body ownership beyond visuotactile synchrony/asynchrony effects^{34,37} ; RHI-related PPC activity is closely linked to the individual visuotactile stimulations delivered to participants' real and rubber hands, whereas premotor activity is more sustained between individual touches and reflects the between IAF and temporal integration continuous feeling of hand ownership"

The final part of the sentence is not 100% clear to me - in which sense premotor activity is more sustained, and what is the result supporting this claim? To me the dissociation between premotor and parietal results are interesting, and can be framed in a more general way, i.e., the dissociations between "higher-lever" experience-related encoding (body ownership) and and lower-lever sensory integration tasks. By the way this rules out also our previous concern about the specificity of the tasks vs. the objection of th participants responding in the same way to any task (our previous point 5)

Reply: We apologize for the lack of clarity in the second part of the sentence. What we intended to convey is that a previous ECoG study by Guterstam et al. (2019) found that activity in the posterior parietal cortex (PPC) closely tracked the timing of individual visuotactile stimulations, whereas premotor cortex (PMC) activity was more sustained and less time-locked to specific touch events. This sustained activity in the PMC was interpreted as reflecting the ongoing feeling of hand ownership rather than the immediate processing of sensory input. We believe this aligns with our current finding that IAF over the PMC correlates specifically with body ownership, but not with the visuotactile simultaneity temporal binding window. This supports the idea that PMC-related oscillatory dynamics may reflect more complex multisensory integrative processes involved in the experience of body ownership rather than the visuotactile integration of individual brief touch events. We have now clarified this point in the revised text and added the appropriate reference to Guterstam et al. (2019) (page 16, line 8):

*"The apparent selectivity of premotor IAF for body ownership aligns well with previous literature showing that activity in this area is related to body ownership beyond visuotactile synchrony/asynchrony effects^{34,37} ; **for instance, Guterstam and colleagues, using ECoG⁴², found** that RHI-related PPC activity is closely linked to the individual visuotactile stimulations delivered to*

participants' real and rubber hands, whereas premotor activity is more sustained between individual touches and reflects the continuous feeling of hand ownership⁴²

Said that, it would be fair to report also an analyses from a control area showing no correlation with the tasks at all to demonstrate specificity of the effects. At least in the supplementary materials.

Reply: We thank the reviewer for this thoughtful suggestion. Our study was designed as a hypothesis-driven test of whether parietal IAF, identified a priori from extensive prior work on visuotactile integration and body ownership, predicts temporal binding windows in both tasks. The aim of the study was therefore not to localise the effect across the cortex, but to test a mechanistic prediction about the posterior parietal cortex specifically. Demonstrating true cortical specificity would require source-level analyses based on individual MRIs, which is beyond the scope of the current manuscript and not necessary to address our central hypothesis. At the sensor level, selecting a “control” electrode risks introducing interpretational ambiguity due to the limited spatial resolution of EEG.

Importantly, we do not claim exclusivity of the PPC effect. We explicitly acknowledge in the Discussion that the observed relationship between IAF and temporal integration may not be exclusive to the PPC (see page 15, lines 32–33). Further exploratory studies using higher spatial resolution techniques (e.g., source reconstruction or MEG) may investigate whole-brain patterns of activity to address questions of anatomical specificity from a more data-driven perspective.

Besides these minor points, the paper is suitable for publications for me.

Once again congratulations to the authors for this excellent study, and thanks for taking into account properly all our comments.

We thank the reviewer for the valuable and constructive feedback that has helped us improve the article.

Andrea Serino